# Logarithmic Regret Bound in Partially Observable Linear Dynamical Systems

**Sahin Lale**
Caltech
alale@caltech.edu

**Kamyar Azizzadenesheli**
Purdue University
kamyar@purdue.edu

**Babak Hassibi**
Caltech
hassibi@caltech.edu

**Anima Anandkumar**
Caltech
anima@caltech.edu

## Abstract

We study the problem of system identification and adaptive control in partially observable linear dynamical systems. Adaptive and closed-loop system identification is a challenging problem due to correlations introduced in data collection. In this paper, we present the first model estimation method with finite-time guarantees in both open and closed-loop system identification. Deploying this estimation method, we propose adaptive control online learning (ADAPTON), an efficient reinforcement learning algorithm that adaptively learns the system dynamics and continuously updates its controller through online learning steps. ADAPTON estimates the model dynamics by occasionally solving a linear regression problem through interactions with the environment. Using policy re-parameterization and the estimated model, ADAPTON constructs counterfactual loss functions to be used for updating the controller through online gradient descent. Over time, ADAPTON improves its model estimates and obtains more accurate gradient updates to improve the controller. We show that ADAPTON achieves a regret upper bound of polylog $(T)$, after $T$ time steps of agent-environment interaction. To the best of our knowledge, ADAPTON is the first algorithm that achieves polylog $(T)$ regret in adaptive control of *unknown* partially observable linear dynamical systems which includes linear quadratic Gaussian (LQG) control.

## 1 Introduction

Reinforcement learning (RL) in unknown partially observable linear dynamical systems with the goal of minimizing a cumulative cost is one of the central problems in adaptive control [1]. In this setting, a desirable RL agent needs to efficiently *explore* the environment to learn the system dynamics, and *exploit* the gathered experiences to minimize overall cost [2]. However, since the underlying states of the systems are not fully observable, learning the system dynamics with finite time guarantees brings substantial challenges, making it a long-lasting problem in adaptive control. In particular, when the latent states of a system are not fully observable, future observations are correlated with the past inputs and observations through the latent states. These correlations are even magnified when closed-loop controllers, those that naturally use past experiences to come up with control inputs, are deployed. Therefore, more sophisticated estimation methods that consider these complicated and unknown correlations are required for learning the dynamics.

In recent years, a series of works have studied this learning problem and presented a range of novel methods with finite-sample learning guarantees. These studies propose to employ i.i.d. Gaussian excitation as the control input, collect system outputs and estimate the model parameters using the data

collected. The use of i.i.d. Gaussian noise as the open-loop control input (not using past experiences) mitigates the correlation in the inputs and the output observations. For stable systems, these methods provide efficient ways to learn the model dynamics with confidence bounds of $\tilde{\mathcal{O}}(1/\sqrt{T})$, after $T$ times step of agent-environment interaction [3–6]. Here $\tilde{\mathcal{O}}(\cdot)$ denotes the order up to logarithmic factors. Deploying i.i.d. Gaussian noise for a long period of time to estimate the model parameters has been the common practice in adaptive control since incorporating closed-loop controller introduces significant challenges to learn the model dynamics [7].

These estimation techniques later have been deployed to propose explore-then-commit based RL algorithms to minimize regret, *i.e.*, how much more cost an agent suffers compared to the cost of a baseline policy [8]. These algorithms deploy i.i.d. Gaussian noise as the control input to learn the model parameters in the *explore* phase and then exploit these estimates during the *commit* phase to minimize regret. Among these works, Lale et al. [9] and Simchowitz et al. [6] respectively propose to use optimism [10] and online convex optimization [11] during the commit phase. These works attain regret of $\tilde{\mathcal{O}}(T^{2/3})$ in the case of convex cost functions. Moreover, in the case of strongly convex cost functions, Mania et al. [12], Simchowitz et al. [6] show that exploiting the strong convexity allows to guarantee regret of $\tilde{\mathcal{O}}(\sqrt{T})$. These methods heavily rely on the lack of correlation achieved by using i.i.d. Gaussian noise as the open-loop control input to estimate the model. Therefore, they do not generalize to the adaptive settings, where the past observations are used to continuously improve both model estimates and the controllers. These challenges pose the following two open problems:

"*Can we estimate the model parameters in closed-loop setting with finite-time guarantees?*"

"*Can we utilize such an estimation method, and propose an RL algorithm to significantly improve regret in partially observable linear dynamical systems?*"

In this paper, we give **affirmative** answers to both of these questions:

• **Novel closed-loop estimation method:** We introduce the first system identification method that allows to estimate the model parameters with finite-time guarantees in both open and closed-loop setting. We exploit the classical predictive form representation of the system that goes back to Kalman [13] and reformulate each output as a linear function of previous control inputs and outputs with an additive i.i.d. Gaussian noise. This reformulation allows to address the limitations of the prior open-loop estimation methods in handling the correlations in inputs and outputs. We state a novel least squares problem to recover the model parameters. We show that when the controllers persistently excite the system, the parameter estimation error is $\tilde{\mathcal{O}}(1/\sqrt{T})$ after $T$ samples. Our method allows updating the model estimates while controlling the system with an adaptive controller.

• **Novel RL algorithm for partially observable linear dynamical systems:** Leveraging our novel estimation method, we propose **adapt**ive control **on**line learning algorithm (ADAPTON) that *adaptively* learns the model dynamics and efficiently uses the model estimates to continuously optimize the controller and reduce the cumulative cost. ADAPTON operates in growing size epochs and in the beginning of each epoch estimates the model parameters using our novel model estimation method. During each epoch, following the online learning procedure introduced by Simchowitz et al. [6], ADAPTON utilizes a convex policy reparameterization of linear controllers and the estimated model dynamics to construct counterfactual loss functions. ADAPTON then deploys online gradient descent on these loss functions to gradually optimize the controller. We show that as the model estimates improve, the gradient updates become more accurate, resulting in improved controllers.

We show that ADAPTON attains a regret upper bound of polylog$(T)$ after $T$ time steps of agent-environment interaction, when the cost functions are strongly convex. To the best of our knowledge, this is the first logarithmic regret bound for partially observable linear dynamical systems with unknown dynamics which includes the canonical LQG setting. The presented regret bound improves $\tilde{\mathcal{O}}(\sqrt{T})$ regret of Simchowitz et al. [6], Mania et al. [12] in stochastic setting with the help of novel estimation method which allows updating model estimates during control (Table 1).

## 2 Preliminaries

We denote the Euclidean norm of a vector $x$ as $\|x\|_2$. For a given matrix $A$, $\|A\|_2$ is its spectral norm, $\|A\|_F$ is its Frobenius norm, $A^\top$ is its transpose, $A^\dagger$ is its Moore-Penrose inverse, and

Table 1: Comparison with prior works in partially observable linear dynamical systems.

| Work | Regret | Cost | Identification | Noise |
|---|---|---|---|---|
| Lale et al. [9] | $T^{2/3}$ | Convex | Open-Loop | Stochastic |
| Simchowitz et al. [6] | $T^{2/3}$ | Convex | Open-Loop | Adversarial |
| Mania et al. [12] | $\sqrt{T}$ | Strongly Convex | Open-Loop | Stochastic |
| Simchowitz et al. [6] | $\sqrt{T}$ | Strongly Convex | Open-Loop | Semi-adversarial |
| **This work** | $\mathrm{polylog}(T)$ | Strongly Convex | Closed-Loop | Stochastic |

$\mathrm{Tr}(A)$ is the trace of $A$. $\rho(A)$ denotes the spectral radius of $A$, *i.e.*, the largest absolute value of its eigenvalues. The j-th singular value of a rank-$n$ matrix $A$ is denoted by $\sigma_j(A)$, where $\sigma_{\max}(A) := \sigma_1(A) \geq \sigma_2(A) \geq \ldots \geq \sigma_n(A) := \sigma_{\min}(A) > 0$. $I$ is the identity matrix with appropriate dimensions. $\mathcal{N}(\mu, \Sigma)$ denotes a multivariate normal distribution with mean vector $\mu$ and covariance matrix $\Sigma$.

**State space form:** Consider an unknown discrete-time linear time-invariant system $\Theta$,

$$x_{t+1} = Ax_t + Bu_t + w_t, \qquad y_t = Cx_t + z_t, \tag{1}$$

where $x_t \in \mathbb{R}^n$ is the (latent) state of the system, $u_t \in \mathbb{R}^p$ is the control input, and the observation $y_t \in \mathbb{R}^m$ is the output of the system. At each time step $t$, the system is at state $x_t$ and the agent observes $y_t$, *i.e.*, an imperfect state information. Then, the agent applies a control input $u_t$, observes the loss function $\ell_t$, pays the cost of $c_t = \ell_t(y_t, u_t)$, and the system evolves to a new $x_{t+1}$ at time step $t + 1$. Let $(\mathcal{F}_t; t \geq 0)$ be the corresponding filtration. For any $t$, conditioned on $\mathcal{F}_{t-1}$, $w_t$ and $z_t$ are $\mathcal{N}(0, \sigma_w^2 I)$ and $\mathcal{N}(0, \sigma_z^2 I)$ respectively. In this paper, in contrast to the standard assumptions in LQG literature that the algorithm is given the knowledge of both $\sigma_w^2$ and $\sigma_z^2$, we only assume the knowledge of their upper and lower bounds, i.e., $\overline{\sigma}_w^2, \underline{\sigma}_w^2, \overline{\sigma}_z^2$, and $\underline{\sigma}_z^2$, such that, $0 < \underline{\sigma}_w^2 \leq \sigma_w^2 \leq \overline{\sigma}_w^2$ and $0 < \underline{\sigma}_z^2 \leq \sigma_z^2 \leq \overline{\sigma}_z^2$, for some finite $\overline{\sigma}_w^2, \overline{\sigma}_z^2$. For the system $\Theta$, let $\Sigma$ be the unique positive semidefinite solution to the following DARE (Discrete Algebraic Riccati Equation),

$$\Sigma = A\Sigma A^\top - A\Sigma C^\top \left(C\Sigma C^\top + \sigma_z^2 I\right)^{-1} C\Sigma A^\top + \sigma_w^2 I. \tag{2}$$

$\Sigma$ can be interpreted as the steady state error covariance matrix of state estimation under $\Theta$.

**Predictor form:** An equivalent and common representation of the system $\Theta$ in (1), is its predictor form representation introduced by Kalman [13] and characterized as,

$$\hat{x}_{t+1} = \bar{A}\hat{x}_t + Bu_t + Fy_t, \qquad y_t = C\hat{x}_t + e_t, \tag{3}$$

where $F = A\Sigma C^\top \left(C\Sigma C^\top + \sigma_z^2 I\right)^{-1}$ is the Kalman filter gain in the observer form, $e_t$ is the zero mean white innovation process and $\bar{A} = A - FC$. In this equivalent representation of system, the state $\hat{x}_t$ can be seen as the estimate of the state in (1).

**Definition 2.1** (Controllability & Observability). *A system is $(A, B)$ controllable if the controllability matrix $[B \ \ AB \ \ A^2B \ldots A^{n-1}B]$ has full row rank. As the dual, a system is $(A, C)$ observable if the observability matrix $[C^\top \ \ (CA)^\top \ \ (CA^2)^\top \ldots (CA^{n-1})^\top]^\top$ has full column rank.*

We assume that the unknown system $\Theta$ is $(A, B)$ controllable, $(A, C)$ observable and $(A, F)$ controllable. This provides exponential convergence of the Kalman filter to the steady-state. Thus, without loss of generality, for the simplicity of analysis, we assume that $x_0 \sim \mathcal{N}(0, \Sigma)$, *i.e.*, the system starts at the steady-state. In the steady state, $e_t \sim \mathcal{N}\left(0, C\Sigma C^\top + \sigma_z^2 I\right)$.

We assume that the unknown system $\Theta$ is order $n$ and $\rho(A) < 1$. Let $\Phi(A) = \sup_{\tau \geq 0} \|A^\tau\| / \rho(A)^\tau$. In the following we consider the standard setting where $\Phi(A)$ is finite. The above mentioned construction is the general setting for the majority of literature on both estimation and regret minimization [3–6, 9, 12] for which the main challenge is the estimation and the controller design.

Note that, recently there has been significant effort to generalize this setting to stabilizable systems when a stabilizing controller is given [6]. However, many partially observable linear dynamical systems cannot be stabilized by a static feedback controller and the assumption of the existence of such controller can be restrictive [14]. Therefore, in this work, we consider the setting described above in order to present the general framework of learning and regret analysis in partially observable linear dynamical systems.

# 3 System Identification

In this section, we first explain the estimation methods that use state-space representation of the system (1) to recover the model parameters and discuss the reason why they cannot provide reliable estimates in closed-loop estimation problems. Then we present our novel estimation method which provides reliable estimates in both open and closed-loop estimation.

**Challenges in using the state-space representation for system identification:** Using the state-space representation in (1), for any positive integer $H$, one can rewrite the output at time $t$ as follows,

$$y_t = \sum_{i=1}^{H} CA^{i-1}Bu_{t-i} + CA^H x_{t-H} + z_t + \sum_{i=0}^{H-1} CA^i w_{t-i-1}. \tag{4}$$

**Definition 3.1** (Markov Parameters). *The set of matrices that maps the inputs to the output in (4) is called Markov parameters of the system $\Theta$. They are the first $H$ parameters of the Markov operator, $\mathbf{G} = \{G^{[i]}\}_{i \geq 0}$ with $G^{[0]} = 0_{m \times p}$, and $\forall i > 0$, $G^{[i]} = CA^{i-1}B$ that uniquely describes the system behavior. Moreover, $\mathbf{G}(H) = [G^{[0]} G^{[1]} \dots G^{[H-1]}] \in \mathbb{R}^{m \times Hp}$ denotes the $H$-length Markov parameters matrix.*

For $\kappa_{\mathbf{G}} \geq 1$, let the Markov operator of $\Theta$ be bounded, *i.e.*, $\sum_{i \geq 0} \|G^{[i]}\| \leq \kappa_{\mathbf{G}}$. Due to stability of $A$, the second term in (4) decays exponentially and for large enough $H$ it becomes negligible. Therefore, using Definition 3.1, we obtain the following for the output at time $t$,

$$y_t \approx \sum_{i=0}^{H} G^{[i]} u_{t-i} + z_t + \sum_{i=0}^{H-1} CA^i w_{t-i-1}. \tag{5}$$

From this formulation, a least squares estimation problem can be formulated using outputs as the dependent variable and the concatenation of $H$ input sequences $\bar{u}_t = [u_t, \dots, u_{t-H}]$ as the regressor to recover the Markov parameters of the system:

$$\widehat{\mathbf{G}}(H) = \arg\min_X \sum_{t=H}^{T} \|y_t - X\bar{u}_t\|_2^2. \tag{6}$$

Prior finite-time system identification algorithms propose to use i.i.d. zero-mean Gaussian noise for the input, to make sure that the two noise terms in (5) are not correlated with the inputs *i.e.* open-loop estimation. This lack of correlation allows them to solve (6), estimate the Markov parameters and develop finite-time estimation error guarantees [3, 4, 9, 15]. From Markov parameter estimates, they recover the system parameters $(A, B, C)$ up to similarity transformation using singular value decomposition based methods like Ho-Kalman algorithm [16].

However, when a controller designs the inputs based on the history of inputs and observations, the inputs become highly correlated with the past process noise sequences, $\{w_i\}_{i=0}^{t-1}$. This correlation prevents consistent and reliable estimation of Markov parameters using (6). Therefore, these prior open-loop estimation methods do not generalize to the systems that adaptive controllers generates the inputs for estimation, *i.e.*, closed-loop estimation. In order to overcome this issue, we exploit the predictor form of the system $\Theta$ and design a novel system identification method that provides consistent and reliable estimates both in closed and open-loop estimation problems.

**Novel estimation method for partially observable linear dynamical systems:** Using the predictor form representation (3), for a positive integer $H_e$, the output at time $t$ can be rewritten as follows,

$$y_t = \sum_{k=0}^{H_e-1} C\bar{A}^k \left(Fy_{t-k-1} + Bu_{t-k-1}\right) + e_t + C\bar{A}^{H_e} x_{t-H_e}. \tag{7}$$

Using the open or closed-loop generated input-output sequences up to time $\tau$, $\{y_t, u_t\}_{t=1}^{\tau}$, we construct subsequences of $H_e$ input-output pairs for $H_e \leq t \leq \tau$,

$$\phi_t = \left[y_{t-1}^{\top} \dots y_{t-H_e}^{\top} u_{t-1}^{\top} \dots u_{t-H_e}^{\top}\right]^{\top} \in \mathbb{R}^{(m+p)H_e}.$$

The output of the system, $y_t$ can be represented using $\phi_t$ as:

$$y_t = \mathcal{G}_{\mathbf{y}} \phi_t + e_t + C\bar{A}^{H_e} x_{t-H_e} \text{ for } \mathcal{G}_{\mathbf{y}} = \left[CF \ C\bar{A}F \ \dots \ C\bar{A}^{H_e-1}F \ CB \ C\bar{A}B \ \dots \ C\bar{A}^{H_e-1}B\right]. \tag{8}$$

Notice that $\bar{A}$ is stable due to $(A, F)$-controllability of $\Theta$ [17]. Therefore, with a similar argument used in (4), for $H_e = O(\log(T))$, the last term in (7) is negligible. This yields into a linear model of the dependent variable $y_t$ and the regressor $\phi_t$ with additive i.i.d. Gaussian noise $e_t$:

$$y_t \approx \mathcal{G}_{\mathbf{y}} \phi_t + e_t. \tag{9}$$

For this model, we achieve consistent and reliable estimates by solving the following regularized least squares problem,

$$\widehat{\mathcal{G}}_{\mathbf{y}} = \arg\min_X \lambda\|X\|_F^2 + \sum_{t=H_e}^{\tau} \|y_t - X\phi_t\|_2^2. \tag{10}$$

This problem does not require any specification on how the inputs are generated and therefore can be deployed in both open and closed-loop estimation problems. Exploiting the specific structure of $\mathcal{G}_{\mathbf{y}}$ in (8), we design a procedure named SYSID, which recovers model parameters from $\widehat{\mathcal{G}}_{\mathbf{y}}$ (see Appendix B for details). To give an overview, SYSID forms two Hankel matrices from the blocks of $\widehat{\mathcal{G}}_{\mathbf{y}}$ which corresponds to the product of observability and controllability matrices. SYSID applies a variant of Ho-Kalman procedure and similar to this classical algorithm, it uses singular value decomposition of these Hankel matrices to recover model parameter estimates $(\hat{A}, \hat{B}, \hat{C})$ and finally constructs $\widehat{\mathbf{G}}(H)$. For the persistently exciting inputs, the following gives the first finite-time system identification guarantee in both open and closed-loop estimation problems (see Appendix D for the proof).

**Theorem 1** (System Identification). *If the inputs are persistently exciting, then for $T$ input-output pairs, as long as $T$ is large enough, solving the least squares problem in (10) and deploying SYSID procedure provides model parameter estimates $(\hat{A}, \hat{B}, \hat{C}, \widehat{\mathbf{G}}(H))$ in which there exists a similarity transformation $\mathbf{S} \in \mathbb{R}^{n \times n}$ such that, with high probability,*

$$\|\hat{A} - \mathbf{S}^{-1}A\mathbf{S}\|, \|\hat{B} - \mathbf{S}^{-1}B\|, \|\hat{C} - C\mathbf{S}\|, \|\widehat{\mathbf{G}}(H) - \mathbf{G}(H)\| = \tilde{\mathcal{O}}(1/\sqrt{T}) \tag{11}$$

## 4 Controller and Regret

In this section we describe the class of linear dynamic controllers (LDC) and show how a convex policy reparameterization can provide accurate approximations of the LDC controllers. Then we provide the regret definition, *i.e.* performance metric, of the adaptive control task.

**Linear dynamic controller (LDC):** An LDC policy $\pi$ is a linear controller with an internal state dynamics of

$$s_{t+1}^{\pi} = A_{\pi}s_t^{\pi} + B_{\pi}y_t, \qquad u_t^{\pi} = C_{\pi}s_t^{\pi} + D_{\pi}y_t, \tag{12}$$

where $s_t^{\pi} \in \mathbb{R}^s$ is the state of the controller, $y_t$ is the input to the controller, *i.e.* observation from the system that controller is designing a policy for, and $u_t^{\pi}$ is the output of the controller. LDC controllers provide a large class of controller. For instance, when the problem is canonical LQG setting, the optimal policy is known to be a LDC policy [1].

**Nature's output:** Using (4), we can further decompose the generative components of $y_t$ to obtain,

$$y_t = z_t + CA^t x_0 + \sum_{i=0}^{t-1} CA^{t-i-1}w_i + \sum_{i=0}^{t} G^{[i]}u_{t-i}$$

Notice that first three components generating $y_t$ are derived from the uncontrollable noise processes in the system, while the last one is a linear combination of control inputs. The first three components are known as Nature's $y$, *i.e.*, Nature's output [6, 18], of the system,

$$b_t(\mathbf{G}) := y_t - \sum_{i=0}^{t-1} G^{[i]}u_{t-i} = z_t + CA^t x_0 + \sum_{i=0}^{t-1} CA^{t-i-1}w_i. \tag{13}$$

The ability to define Nature's $y$ is a unique characteristics of linear dynamical systems. At any time step $t$, after following a sequence of control inputs $\{u_i\}_{i=0}^{t}$, and observing $y_t$, we can compute $b_t(\mathbf{G})$ using (13). This quantity allows for counterfactual reasoning about the outcome of the system. Particularly, having access to $\{b_{\tau-t}(\mathbf{G})\}_{t\geq 0}$, we can reason what the outputs $y'_{\tau-t}$ of the system would have been, if the agent, instead, had taken other sequence of control inputs $\{u'_i\}_{i=0}^{\tau-t}$, *i.e.*,

$$y'_{\tau-t} = b_{\tau-t}(\mathbf{G}) + \sum_{i=0}^{\tau-t-1} G^{[i]}u'_{\tau-t-i}.$$

This property indicates that we can use $\{b_{\tau-t}(\mathbf{G})\}_{t\geq 0}$ to evaluate the quality of any other potential input sequences, and build a desirable controller, as elaborated in the following.

**Disturbance feedback control (DFC):** In this work, we adopt a convex policy reparametrization called DFC introduced by Simchowitz et al. [6]. A DFC policy of length $H'$ is defined as a set of parameters, $\mathbf{M}(H') := \{M^{[i]}\}_{i=0}^{H'-1}$, prescribing the control input of

$$u_t^{\mathbf{M}} = \sum_{i=0}^{H'-1} M^{[i]}b_{t-i}(\mathbf{G}). \tag{14}$$

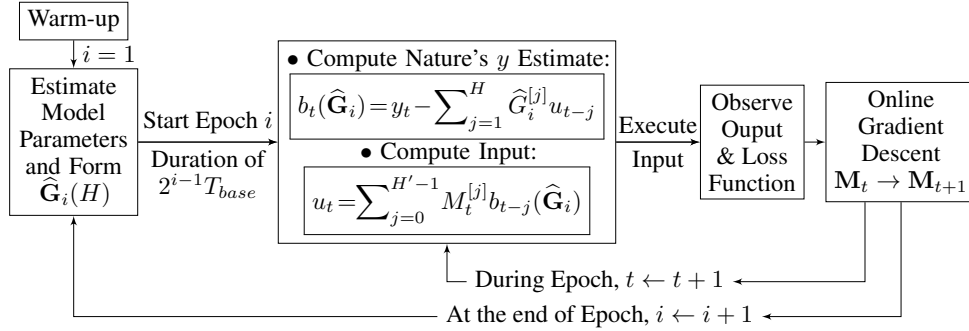

Figure 1: ADAPTON

for Nature's $y$, $\{b_{t-i}(\mathbf{G})\}_{i=0}^{H'-1}$. The DFC policy construction is in parallel with the classical Youla parametrization [18] which states that any linear controller can be prescribed as acting on past noise sequences. Thus, DFC policies can be regarded as truncated approximations to LDCs. More formally, any stabilizing LDC policy can be well-approximated as a DFC policy (see Appendix A).

Define the convex compact sets of DFCs, $\mathcal{M}_\psi$ and $\mathcal{M}$ such that all the controllers in these sets persistently excite the system $\Theta$. The precise definition of persistence of excitation condition is given in Appendix E.2. The persistence of excitation condition is a mild condition and it only requires a significantly wide matrix that maps past $H'$ noise sequences to input to be full row rank. This condition holds in many well-known controllers such as $H_2, H_\infty$. Moreover, if a controller is persistently exciting there exists a neighborhood around it consisting of persistently exciting controllers.

The controllers $\mathbf{M}(H'_0) \in \mathcal{M}_\psi$ are bounded *i.e.*, $\sum_{i\geq 0}^{H'_0-1} \|M^{[i]}\| \leq \kappa_\psi$ and $\mathcal{M}$ is an $r$-expansion of $\mathcal{M}_\psi$, *i.e.*, $\mathcal{M} = \{\mathbf{M}(H') = \mathbf{M}(H'_0) + \Delta : \mathbf{M}(H'_0) \in \mathcal{M}_\psi, \sum_{i\geq 0}^{H'-1} \|\Delta^{[i]}\| \leq r\kappa_\psi\}$ where $H'_0 = \lfloor \frac{H'}{2} \rfloor - H$. Thus, all controllers $\mathbf{M}(H') \in \mathcal{M}$ are also bounded $\sum_{i\geq 0}^{H'-1} \|M^{[i]}\| \leq \kappa_\mathcal{M}$ where $\kappa_\mathcal{M} = \kappa_\psi(1 + r)$. Throughout the interaction with the system, the agent has access to $\mathcal{M}$.

**Loss function:** The loss function at time $t$, $\ell_t$, is smooth and strongly convex for all $t$, i.e., $0 \prec \underline{\alpha}_{loss}I \preceq \nabla^2\ell_t(\cdot,\cdot) \preceq \overline{\alpha}_{loss}I$ for a finite constant $\overline{\alpha}_{loss}$. Note that the standard quadratic regulatory costs of $\ell_t(y_t, u_t) = y_t^\top Q_t y_t + u_t^\top R_t u_t^\top$ with bounded positive definite matrices $Q_t$ and $R_t$ are special cases of the mentioned setting. For all $t$, the unknown lost function $c_t = \ell_t(\cdot,\cdot)$ is non-negative strongly convex and associated with a parameter $L$, such that for any $R$ with $\|u\|, \|u'\|, \|y\|, \|y'\| \leq R$, we have,

$$|\ell_t(y,u) - \ell_t(y',u')| \leq LR(\|y - y'\| + \|u - u'\|) \quad \text{and} \quad |\ell_t(y,u)| \leq LR^2. \qquad (15)$$

**Regret definition:** We evaluate the agent's performance by its regret with respect to $\mathbf{M}_\star$, the optimal, in hindsight, DFC policy in the given set $\mathcal{M}_\psi$, *i.e.*, $\mathbf{M}_\star = \arg\min_{\mathbf{M}\in\mathcal{M}_\psi} \sum_{t=1}^T \ell_t(y_t^\mathbf{M}, u_t^\mathbf{M})$. After $T$ step of interaction, the agent's regret is denoted as

$$\text{REGRET}(T) = \sum_{t=1}^T c_t - \ell_t(y^{\mathbf{M}_\star}, u^{\mathbf{M}_\star}). \qquad (16)$$

## 5  ADAPTON

In this section, we present ADAPTON, a sample efficient **adapt**ive control **on**line learning algorithm which learns the model dynamics through interaction with the environment and continuously deploys online convex optimization to improve the control policy. ADAPTON is illustrated in Figure 1 and the detailed pseudo-code is provided in Appendix C.

**Warm-up:** ADAPTON starts with a fixed warm-up period and applies $u_t \sim \mathcal{N}(0, \sigma_u^2 I)$ for the first $T_w$ time steps. The length of the warm-up period is chosen to guarantee an accountable first estimate of the system, the persistence of excitation during the adaptive control period and the stability of the online learning algorithm on the underlying system.

**Adaptive control in epochs:** After warm-up, ADAPTON starts controlling the system and operates in epochs with doubling length. ADAPTON sets the base period $T_{base}$ to the initial value $T_{base} = T_w$ and for each epoch $i$, it runs for $2^{i-1}T_{base}$ time steps.

**Model estimation in the beginning of epochs:** At the beginning of each epoch $i$, ADAPTON exploits the past experiences up to epoch $i$. It deploys the proposed closed-loop estimation method and solves (10) to estimate $\mathcal{G}_\mathbf{y}$. ADAPTON then exploits the construction of true $\mathcal{G}_\mathbf{y}$ to estimate model parameter estimates $\hat{A}_i, \hat{B}_i, \hat{C}_i$ and constructs an estimate of $H$-length Markov parameters matrix, $\widehat{\mathbf{G}}_i(H)$, via SYSID described in Section 3 and provided in Appendix B.

**Control input, output and loss during the epochs:** ADAPTON utilizes $\widehat{\mathbf{G}}_i(H)$ and the past inputs to estimate the Nature's outputs, $b_t(\widehat{\mathbf{G}}_i) = y_t - \sum_{j=1}^{H} \widehat{G}_i^{[j]} u_{t-j}$. Using these estimates, ADAPTON executes a DFC policy $\mathbf{M}_t \in \mathcal{M}$ such that $u_t^{\mathbf{M}_t} = \sum_{j=0}^{H'-1} M_t^{[j]} b_{t-j}(\widehat{\mathbf{G}}_i)$ and observes the output of $y_t^{\mathbf{M}_t}$. Finally, ADAPTON receives the loss function $\ell_t$, pays a cost of $\ell(y_t^{\mathbf{M}_t}, u_t^{\mathbf{M}_t})$.

**Counterfactual input, output, loss:** ADAPTON uses counterfactual reasoning introduced in Simchowitz et al. [6] to update its controller. After observing the loss function $\ell_t$, it constructs,

$$\tilde{u}_{t-j}(\mathbf{M}_t, \widehat{\mathbf{G}}_i) = \sum_{l=0}^{H'-1} M_t^{[l]} b_{t-j-l}(\widehat{\mathbf{G}}_i), \tag{17}$$

the counterfactual inputs, which are the recomputations of past inputs as if the current DFC policy is applied using Nature's $y$ estimates. Then, ADAPTON reasons about the counterfactual output of the system. Using the current Nature's $y$ estimate and the counterfactual inputs, the agent approximates what the output of the system could be, if counterfactual inputs had been applied,

$$\tilde{y}_t(\mathbf{M}_t, \widehat{\mathbf{G}}_i) = b_t(\widehat{\mathbf{G}}_i) + \sum_{j=1}^{H} \hat{G}_i^{[j]} \tilde{u}_{t-j}(\mathbf{M}_t, \widehat{\mathbf{G}}_i). \tag{18}$$

Using the counterfactual inputs, output and the revealed loss function $\ell_t$, ADAPTON finally constructs,

$$f_t(\mathbf{M}_t, \widehat{\mathbf{G}}_i) = \ell_t(\tilde{y}_t(\mathbf{M}_t, \widehat{\mathbf{G}}_i), \tilde{u}_t(\mathbf{M}_t, \widehat{\mathbf{G}}_i)). \tag{19}$$

which is termed as the counterfactual loss. It is ADAPTON's approximation of what the cost would have been at time $t$, if the current DFC policy was applied until time $t$. It gives a performance evaluation of the current DFC policy to ADAPTON for updating the policy. Note that the Markov parameter estimates are crucial in the accuracy of this performance evaluation.

**Online convex optimization:** In order to optimize the controller during the epoch, at each time step, ADAPTON runs online gradient descent on the counterfactual loss function $f_t(\mathbf{M}_t, \widehat{\mathbf{G}}_i)$ while keeping the updates in the set $\mathcal{M}$ via projection [6],

$$\mathbf{M}_{t+1} = proj_\mathcal{M} \left( \mathbf{M}_t - \eta_t \nabla_\mathbf{M} f_t \left( \mathbf{M}, \widehat{\mathbf{G}}_i \right) \Big|_{\mathbf{M}_t} \right).$$

Notice that if ADAPTON had access to the underlying Markov operator $\mathbf{G}$, the counterfactual loss would have been the true loss of applying the current DFC policy until time $t$, up to truncation. By knowing the exact performance of the DFC policy, online gradient descent would have obtained accurate updates. Using the counterfactual loss for optimizing the controller causes an error in the gradient updates which is a function of estimation error of $\widehat{\mathbf{G}}_i$. Therefore, as the Markov estimates improve via our closed-loop estimation method, the gradient updates get more and more accurate.

## 6 Regret Analysis

In this section, we first provide the closed-loop learning guarantee of ADAPTON, then show that ADAPTON maintains stable system dynamics and present the regret upper bound for ADAPTON.

**Closed-loop learning guarantee:** In the beginning of adaptive control epochs, ADAPTON guarantees that Markov parameter estimates are accurate enough that deploying any controller from set $\mathcal{M}$ provides persistence of excitation in inputs (see Appendix E). Under this guarantee, using our novel estimation method at the beginning of any epoch $i$ ensures that during the epoch, $\|\widehat{\mathbf{G}}_i(H) - \mathbf{G}(H)\| = \tilde{\mathcal{O}}(1/\sqrt{2^{i-1}T_{base}})$, due to Theorem 1 and doubling length epochs.

**Stable system dynamics with ADAPTON:** Since $w_t$ and $z_t$ are Gaussian disturbances, from standard concentration results we have that Nature's $y$ is bounded with high probability for all $t$ (see Appendix F). Thus, let $\|b_t(\mathbf{G})\| \leq \kappa_b$ for some $\kappa_b$. The following lemma shows that during ADAPTON, Markov parameter estimates are well-refined such that the inputs, outputs and the Nature's $y$ estimates of ADAPTON are uniformly bounded with high probability. The proof is in Appendix F.

**Lemma 6.1.** *For all $t$ during the adaptive control epochs, $\|u_t\| \leq \kappa_{\mathcal{M}}\kappa_b$, $\|y_t\| \leq \kappa_b(1 + \kappa_{\mathbf{G}}\kappa_{\mathcal{M}})$ and $\|b_t(\widehat{\mathbf{G}})\| \leq 2\kappa_b$ with high probability.*

**Regret upper bound of ADAPTON:** The regret decomposition of ADAPTON includes 3 main pieces: $(R_1)$ Regret due to warm-up, $(R_2)$ Regret due to online learning controller, $(R_3)$ Regret due to lack of system dynamics knowledge (see Appendix G for exact expressions and proofs). $R_1$ gives constant regret for the short warm-up period. $R_2$ results in $\mathcal{O}(\log(T))$ regret. Note that this regret decomposition and these results follow and adapt Theorem 5 of Simchowitz et al. [6]. The key difference is in $R_3$, which scales quadratically with the Markov parameter estimation error $\|\widehat{\mathbf{G}}_i(H) - \mathbf{G}(H)\|$. Simchowitz et al. [6] deploys open-loop estimation and does not update the model parameter estimates during adaptive control and attains $R_3 = \tilde{\mathcal{O}}(\sqrt{T})$ which dominates the regret upper bound. However, using our novel system identification method with the closed-loop learning guarantees of Markov parameters and the doubling epoch lengths ADAPTON gets $R_3 = \mathcal{O}(\text{polylog}(T))$.

**Theorem 2.** *Given $\mathcal{M}$, a closed, compact and convex set of DFC policies with persistence of excitation, with high probability, ADAPTON achieves logarithmic regret, i.e., REGRET$(T) = polylog(T)$.*

In minimizing the regret, ADAPTON competes against the best DFC policy in the given set $\mathcal{M}$. Recall that any stabilizing LDC policy can be well-approximated as a DFC policy. Therefore, for any LDC policy $\pi$ whose DFC approximation lives in the given $\mathcal{M}$, Theorem 2 can be extended to achieve the first logarithmic regret in LQG setting.

**Corollary 6.1.** *Let $\pi_\star$ be the optimal linear controller for LQG setting. If the DFC approximation of $\pi_\star$ is in $\mathcal{M}_\psi$, then the regret of ADAPTON with respect to $\pi_\star$ is $\sum_{t=1}^{T} c_t - \ell_t(y_t^{\pi_\star}, u_t^{\pi_\star}) = polylog(T)$.*

Note that without any estimation updates during the adaptive control, ADAPTON reduces to a variant of the algorithm given in Simchowitz et al. [6]. While the update rule in ADAPTON results in $\mathcal{O}(\log(T))$ updates in adaptive control period, one can follow different update schemes as long as ADAPTON obtains enough samples in the beginning of the adaptive control period to obtain persistence of excitation. The following is an immediate corollary of Theorem 2 which considers the case when number of epochs or estimations are limited during the adaptive control period.

**Corollary 6.2.** *If enough samples are gathered in the adaptive control period, ADAPTON with any update scheme less than $\log(T)$ updates has REGRET$(T) \in [polylog(T), \tilde{O}(\sqrt{T})]$.*

## 7 Related Works

**Classical results in system identification:** The classical open or closed-loop system identification methods either consider the asymptotic behavior of the estimators or demonstrate the positive and negative empirical performances without theoretical guarantees [19–26]. Most of prior work exploits the state-space form or the innovations form representation of the system. Among all closed-loop estimation methods only a handful consider the predictor form representation for system identification [27, 28]. For an extensive overview of classical system identification is provided in Qin [7].

**Finite-time system identification for partially observable linear dynamical systems:** In partially observable linear systems, most of the prior works focus on open-loop system identification guarantees [3–5, 15, 29–31]. Among all, only Lee and Lamperski [29] considers finite-time closed-loop system identification. However, they analyze the output estimation error rather than explicitly recovering the model parameters as presented in this work. Another body of work aims to extend the problem of estimation and prediction to online convex optimization where a set of strong guarantees on cumulative prediction errors are provided [32–37]

**Regret in fully observable linear dynamical systems:** Efforts in regret analysis of adaptive control in fully observable linear dynamical systems is initiated by seminal work of Abbasi-Yadkori and Szepesvári [38]. They present $\tilde{\mathcal{O}}(\sqrt{T})$ regret upper bound for linear quadratic regulators (LQR)

which are fully observable counterparts of LQG. This work sparked the flurry of research with different directions in the regret analysis in LQRs [39–44, 12, 45]. Recently, Cassel et al. [46] show that logarithmic regret is achievable if only $A$ or $B$ is unknown, and Simchowitz and Foster [47] provide $\tilde{\mathcal{O}}(\sqrt{T})$ regret lower bound for LQR setting with fully unknown systems. However, due to the persistent noise in the observations of the hidden states, the mentioned lower bound does not carry to the partially observable linear dynamical systems.

**Regret in adversarial noise setting:** In adversarial noise setting, most of the works consider full information of the underlying system [48–51]. Recent efforts extend to adaptive control in the adversarial setting for the unknown system model [52, 6].

# 8   Conclusion

In this paper, we propose the first system identification algorithm that provides consistent and reliable estimates with finite-time guarantees in both open and closed-loop estimation problems in partially observable linear dynamical systems. We believe this system identification algorithm fills an important gap in learning linear dynamical systems and is of independent interest for reinforcement learning and control communities. We deploy this estimation technique in ADAPTON, a novel adaptive control algorithm that efficiently learns the model parameters of the underlying dynamical system and deploys projected online gradient descent to design a controller. We show that in the presence of convex set of persistently exciting linear controllers and strongly convex loss functions, ADAPTON achieves a regret upper bound that is polylogarithmic in number of agent-environment interactions. The unique nature of ADAPTON which combines occasional model estimation with continual online convex optimization allows the agent to achieve significantly improved regret in the challenging setting of adaptive control in partially observable linear dynamical systems.

In future work, it would be an interesting direction to further relax the persistence of excitation condition and provide regret analysis in such settings. Moreover, adapting these techniques to adaptive control of non-linear systems or safety constrained control would be other important directions.

## 9 Broader Impact

In this work, we study the two open problems regarding the system identification and the adaptive control in partially observable linear dynamical systems. In the system identification front, we provide the first estimation method that provides finite-time guarantees in estimating the model parameters from the data collected by using a controller that acts based on history of inputs and outputs (closed-loop control). We believe this result is crucial in both theory and practice fronts. It opens doors for developing efficient interactive learning algorithms, that adaptively utilize past experiences to improve performance. We provide our estimation method, using a different representation of the system. We believe the idea of using different representations would inspire new advancements in future methods in system identification.

Moreover, this result provides a solution with theoretical guarantees to a practical problem. In real-world system identification problems, the system is not usually stable and a stabilizing controller is required for data collection to avoid catastrophic results in data collection, e.g. a robot learning to accomplish a task using a stabilizing controller. Prior methods cannot provide finite-sample guarantees in learning dynamics of this setting. However, our novel method provides guarantees in this setting and we think this will be useful in designing policies in many similar RL tasks.

In adaptive control front, we deploy our novel estimation method in an RL algorithm and show that how it can significantly improve the performance. We believe the structure of our algorithm can inspire new developments of RL algorithms in high dimensional and realistic environment when the whole system is not fully observable by the decision making agent.

## Acknowledgments and Disclosure of Funding

S. Lale is supported in part by DARPA PAI. K. Azizzadenesheli gratefully acknowledge the financial support of Raytheon and Amazon Web Services. B. Hassibi is supported in part by the National Science Foundation under grants CNS-0932428, CCF-1018927, CCF-1423663 and CCF-1409204, by a grant from Qualcomm Inc., by NASA's Jet Propulsion Laboratory through the President and Director's Fund, and by King Abdullah University of Science and Technology. A. Anandkumar is supported in part by Bren endowed chair, DARPA PAIHR00111890035 and LwLL grants, Raytheon, Microsoft, Google, and Adobe faculty fellowships.

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
