[Supplementary Material]

# Appendix

In the beginning of this Appendix, we will provide the overall organization of the Appendix and notation table for the paper. Then we will include description of lower bound on the warm-up duration and briefly comment on their goal in helping to achieve the regret result.

**Appendix organization and notation:** In Appendix A, we revisit the precise definitions of LDC and DFC policies and introduce the technical properties that will be used in the proofs. We also provide Lemma A.1 that shows that any stabilizing LDC policy can be well-approximated by an DFC policy. In Appendix B, we provide the details of SYSID procedure that recovers model parameter estimates from the estimate of $\mathcal{G}_\mathbf{y}$ obtained by (10). Appendix C provides the detailed pseudocode of ADAPTON. In Appendix D, we provide the proof Theorem 1 step by step. In Appendix E, we provide the persistence of excitation guarantees for ADAPTON which enables us to achieve consistent estimates by using the new system identification method. In particular, in Appendix E.1 we show the persistence of excitation during the warm-up, in Appendix E.2 we formally define the persistence of excitation property of the controllers in $\mathcal{M}$, *i.e.* (43), and finally in Appendix E.3, we show that the control policies of ADAPTON achieve persistence of excitation during the adaptive control. Appendix F shows that execution of ADAPTON results in stable system dynamics. In Appendix G, we state the formal regret result of the paper, Theorem 5 and provides its proof. Appendix H briefly looks into the case where the loss functions are convex. Finally, in Appendix I, we provide the supporting technical theorems and lemmas. Table 2 provides the notations used throughout the paper.

**Warm-up duration:** The duration of warm-up is chosen as $T_w \geq T_{\max}$, where,

$$T_{\max} := \max\{H, H', T_o, T_A, T_B, T_c, T_{\epsilon_G}, T_{cl}, T_{cx}, T_r\}. \tag{20}$$

This duration guarantees an accountable first estimate of the system ($T_o$, see Appendix E.1), the stability of the online learning algorithm on the underlying system ($T_A, T_B$, see Appendix D), the stability of the inputs and outputs ($T_{\epsilon_\mathbf{G}}$, see Appendix E.3), the persistence of excitation during the adaptive control period ($T_{cl}$, see Appendix E.3), an accountable estimate at the first epoch of adaptive control ($T_c$, see Appendix E.3), the conditional strong convexity of expected counterfactual losses ($T_{cx}$, see Appendix F.1) and the existence of a good comparator DFC policy in $\mathcal{M}$ ($T_r$, see Appendix F.1) . The precise expressions are given throughout the Appendix in stated sections.

Table 2: Useful Notations for the Analysis

| System Not. | Definition |
| --- | --- |
| $\Theta$ | Unknown discrete-time linear time invariant system with dynamics of (1) |
| $x_t$ | Unobserved (latent) state |
| $y_t$ | Observed output |
| $u_t$ | Input to the system |
| $w_t$ | Process Noise, $w_t \sim \mathcal{N}(0, \sigma_w^2 I)$ |
| $z_t$ | Measurement noise, $z_t \sim \mathcal{N}(0, \sigma_z^2 I)$ |
| $\ell_t(y_t, u_t)$ | Revealed loss function after taking action $u_t$ |
| $\Sigma$ | Steady state error covariance matrix of state estimation under $\Theta$ |
| $F$ | Kalman filter gain in the observer form, $F = A\Sigma C^\top (C\Sigma C^\top + \sigma_z^2 I)^{-1}$ |
| $e_t$ | Innovation process, $e_t \sim \mathcal{N}(0, C\Sigma C^\top + \sigma_z^2 I)$ |
| $\Phi(A)$ | Growth rate of powers of $A$, $\Phi(A) = \sup_{\tau \geq 0} \|A^\tau\| / \rho(A)^\tau$ |
| $\mathbf{G}$ | Markov operator, $\mathbf{G} = \{G^{[i]}\}_{i \geq 0}$ with $G^{[0]} = 0$, and $\forall i > 0, G^{[i]} = CA^{i-1}B$ |
| $\mathcal{G}_\mathbf{y}$ | System parameter to be estimated |
| $\phi_t$ | Regressor in the estimation, concatenation of $H_e$ input-output pairs |
| $b_t(\mathbf{G})$ | Nature's $y$, $b_t(\mathbf{G}) := y_t - \sum_{i=0}^{t-1} G^{[i]} u_{t-i}$ |
| $\mathcal{G}^{ol}$ | Operator that maps history of noise and open-loop inputs to $\phi$ |

| Policy Not. | |
| --- | --- |
| $\pi$ | LDC policy |
| $\psi$ | Proper decay function, nonincreasing, $\lim_{h' \to \infty} \psi(h') = 0$ |
| $\Pi(\psi)$ | LDC class with decay function $\psi$, $\forall \pi \in \Pi(\psi)$, $\sum_{i \geq h} \|{G'_\pi}^{[i]}\| \leq \psi(h), \forall h$ |
| $\mathbf{G}'_\pi$ | Markov operator of induced closed-loop system by $\pi$ |
| $\mathbf{M}(H')$ | DFC policy of length $H'$, $\mathbf{M}(H') := \{M^{[i]}\}_{i=0}^{H'-1}$, |
| $u_t^\mathbf{M}$ | Input designed by DFC policy $u_t^\mathbf{M} = \sum_{i=0}^{H'-1} M^{[i]} b_{t-i}(\mathbf{G})$ |
| $\mathcal{M}$ | Given set of persistently exciting DFC controllers |
| $\mathbf{M}_\star$ | Optimal policy in hindsight in $\mathcal{M}$ |
| $\tilde{u}_{t-j}(\mathbf{M}_t, \widehat{\mathbf{G}}_i)$ | Counterfactual input, $\sum_{l=0}^{H'-1} M_t^{[l]} b_{t-j-l}(\widehat{\mathbf{G}}_i)$ |
| $\tilde{y}_{t-j}(\mathbf{M}_t, \widehat{\mathbf{G}}_i)$ | Counterfactual output, $b_t(\widehat{\mathbf{G}}_i) + \sum_{j=1}^{H} \hat{G}_i^{[j]} \tilde{u}_{t-j}(\mathbf{M}_t, \widehat{\mathbf{G}}_i)$ |
| $f_t(\mathbf{M}_t, \widehat{\mathbf{G}}_i)$ | Counterfactual loss, $\ell_t(\tilde{y}_t(\mathbf{M}_t, \widehat{\mathbf{G}}_i), \tilde{u}_t(\mathbf{M}_t, \widehat{\mathbf{G}}_i))$ |

| Quantities | |
| --- | --- |
| $H$ | Length of Markov parameter matrix, $\mathbf{G}(H)$ |
| $H_e$ | Length of estimated operator $\mathcal{G}_\mathbf{y}$ |
| $H'$ | Length of DFC policy |
| $\kappa_\mathbf{G}$ | Bound on the Markov operator, $\sum_{i \geq 0} \|G^{[i]}\| \leq \kappa_\mathbf{G}$ |
| $\kappa_b$ | Bound on Nature's $y$, $\|b_t(\mathbf{G})\| \leq \kappa_b$ |
| $\kappa_u$ | Bound on input $\|u_t\|$ |
| $\kappa_y$ | Bound on output $\|y_t\|$ |
| $\kappa_\mathcal{M}$ | Bound on the DFC policy, $\sum_{i \geq 0}^{H'-1} \|M^{[i]}\| \leq \kappa_\mathcal{M} = (1+r)\kappa_\psi$ |
| $\kappa_\psi$ | Maximum of decay function, $\psi(0)$ |
| $\psi_\mathbf{G}(h)$ | Induced decay function on $\mathbf{G}$, $\sum_{i \geq h} \|G^{[i]}\|$ |
| $T_{base} = T_w$ | Base length of first adaptive control epoch and warm-up duration |

| Estimates | |
| --- | --- |
| $\widehat{\mathcal{G}}_{\mathbf{y},\mathbf{i}}$ | Estimate of $\mathcal{G}_\mathbf{y}$ at epoch $i$ |
| $\widehat{\mathbf{G}}(H)$ | Estimate of Markov parameter matrix |
| $b_t(\widehat{\mathbf{G}}_i)$ | Estimate of Nature's $y$, |
| $\epsilon_\mathbf{G}(i, \delta)$ | Estimation error of Markov operator estimate at epoch $i$, $\tilde{\mathcal{O}}(1/\sqrt{2^{i-1}T_{base}})$ |

# A  Details on LDC Policies and DFC Policies

**Linear Dynamic Controller (LDC):** An LDC policy, $\pi$, is a $s$ dimensional linear controller on a state $s_t^\pi \in \mathbb{R}^s$ of a linear dynamical system $(A_\pi, B_\pi, C_\pi, D_\pi)$, with input $y_t^\pi$ and output $u_t^\pi$, where the state dynamics and the controller evolves as follows,

$$s_{t+1}^\pi = A_\pi s_t^\pi + B_\pi y_t^\pi, \qquad u_t^\pi = C_\pi s_t^\pi + D_\pi y_t^\pi. \tag{21}$$

Deploying a LDC policy $\pi$ on the system $\Theta = (A, B, C)$ induces the following joint dynamics of the $x_t^\pi, s_t^\pi$ and the observation-action process:

$$\begin{bmatrix} x_{t+1}^\pi \\ s_{t+1}^\pi \end{bmatrix} = \underbrace{\begin{bmatrix} A+BD_\pi C & BC_\pi \\ B_\pi C & A_\pi \end{bmatrix}}_{A_\pi'} \begin{bmatrix} x_t^\pi \\ s_t^\pi \end{bmatrix} + \underbrace{\begin{bmatrix} I_n & BD_\pi \\ 0_{s\times n} & B_\pi \end{bmatrix}}_{B_\pi'} \begin{bmatrix} w_t \\ z_t \end{bmatrix}, \quad \begin{bmatrix} y_t^\pi \\ u_t^\pi \end{bmatrix} = \underbrace{\begin{bmatrix} C & 0_{s\times d} \\ D_\pi C & C_\pi \end{bmatrix}}_{C_\pi'} \begin{bmatrix} x_t^\pi \\ s_t^\pi \end{bmatrix} + \underbrace{\begin{bmatrix} 0_{d\times n} & I_d \\ 0_{m\times n} & D_\pi \end{bmatrix}}_{D_\pi'} \begin{bmatrix} w_t \\ z_t \end{bmatrix}$$

where $(A_\pi', B_\pi', C_\pi', D_\pi')$ are the associated parameters of induced closed-loop system. We define the Markov operator for the system $(A_\pi', B_\pi', C_\pi', D_\pi')$, as $\mathbf{G}_\pi' = \{G_\pi'^{[i]}\}_{i=0}$, where $G_\pi'^{[0]} = D_\pi'$, and $\forall i > 0, G_\pi'^{[i]} = C_\pi' A_\pi'^{i-1} B_\pi'$. Let $B_{\pi,z}' := [D_\pi^\top B^\top \quad B_\pi^\top]^\top$ and $C_{\pi,u}' := [D_\pi C \quad C_\pi]$.

**Proper Decay Function:** Let $\psi : \mathbb{N} \to \mathbb{R}_{\geq 0}$ be a proper decay function, such that $\psi$ is non-increasing and $\lim_{h'\to\infty} \psi(h') = 0$. For a Markov operator $\mathbf{G}$, $\psi_\mathbf{G}(h)$ defines the induced decay function on $\mathbf{G}$, i.e., $\psi_\mathbf{G}(h) := \sum_{i\geq h} \|G^{[i]}\|$. $\Pi(\psi)$ denotes the class of LDC policies associated with a proper decay function $\psi$, such that for all $\pi \in \Pi(\psi)$, and all $h \geq 0$, $\sum_{i\geq h} \|G_\pi'^{[i]}\| \leq \psi(h)$. Let $\kappa_\psi := \psi(0)$.

**DFC Policy:** A DFC policy of length $H'$ is defined with a set of parameters, $\mathbf{M}(H') := \{M^{[i]}\}_{i=0}^{H'-1}$, prescribing the control input of $u_t^\mathbf{M} = \sum_{i=0}^{H'-1} M^{[i]} b_{t-i}(\mathbf{G})$, and resulting in state $x_{t+1}^\mathbf{M}$ and observation $y_{t+1}^\mathbf{M}$. In the following, directly using the analysis in Simchowitz et al. [6], we show that for any $\pi \in \Pi(\psi)$ and any input $u_t^\pi$ at time step $t$, there is a set of parameters $\mathbf{M}(H')$, such that $u_t^\mathbf{M}$ is sufficiently close to $u_t^\pi$, and the resulting $y_t^\pi$ is sufficiently close to $y_t^\mathbf{M}$.

**Lemma A.1.** *Suppose $\|b_t(\mathbf{G})\| \leq \kappa_b$ for all $t \leq T$ for some $\kappa_b$. For any LDC policy $\pi \in \Pi(\psi)$, there exist a $H'$ length DFC policy $\mathbf{M}(H')$ such that $\|u_t^\pi - u_t^\mathbf{M}\| \leq \psi(H')\kappa_b$, and $\|y_t^\pi - y_t^\mathbf{M}\| \leq \psi(H')\kappa_\mathbf{G}\kappa_b$. One of the DFC policies that satisfy this is $M^{[0]} = D_\pi$, and $M^{[i]} = C_{\pi,u}' A_\pi'^{i-1} B_{\pi,z}'$ for $0 < i < H'$.*

*Proof.* Let $B_{\pi,w}' := [I_{n\times n}^\top \quad 0_{s\times n}^\top]^\top$, and $B_{\pi,z}' := [D_\pi^\top B^\top \quad B_\pi^\top]^\top$, the columns of $B_\pi'$ applied on process noise, and measurement noise respectively. Similarly $C_{\pi,y}' := [C \quad 0_{s\times d}]$ and $C_{\pi,u}' := [D_\pi C \quad C_\pi]$ are rows of $C_\pi'$ generating the observation and action.

Rolling out the dynamical system defining a policy $\pi$ in (21), we can restate the action $u_t^\pi$ as follows,

$$u_t^\pi = D_\pi z_t + \sum_{i=1}^{t-1} C_{\pi,u}' A_\pi'^{i-1} B_{\pi,z}' z_{t-i} + \sum_{i=1}^{t-1} C_{\pi,u}' A_\pi'^{i-1} B_{\pi,w}' w_{t-i}$$

$$= D_\pi z_t + \sum_{i=1}^{t-1} C_{\pi,u}' A_\pi'^{i-1} B_{\pi,z}' z_{t-i} + C_{\pi,u}' B_{\pi,w}' w_{t-1} + \sum_{i=2}^{t-1} C_{\pi,u}' A_\pi'^{i-1} B_{\pi,w}' w_{t-i}$$

$$= D_\pi z_t + \sum_{i=1}^{t-1} C_{\pi,u}' A_\pi'^{i-1} B_{\pi,z}' z_{t-i} + D_\pi C w_{t-1} + \sum_{i=2}^{t-1} C_{\pi,u}' A_\pi'^{i-1} B_{\pi,w}' w_{t-i}$$

Note that $A_\pi' B_{\pi,w}'$ is equal to $\begin{bmatrix} A + BD_\pi C \\ B_\pi C \end{bmatrix}$. Based on the definition of $A_\pi'$ in (21), we restate $A_\pi'$ as follows,

$$A_\pi' = \begin{bmatrix} A+BD_\pi C & BC_\pi \\ B_\pi C & A_\pi \end{bmatrix} = \begin{bmatrix} BD_\pi C & BC_\pi \\ B_\pi C & A_\pi \end{bmatrix} + \begin{bmatrix} A & 0_{n\times s} \\ 0_{s\times n} & 0_{s\times s} \end{bmatrix}$$

For any given bounded matrices $A'_\pi$ and $A$, and any integer $i > 0$, we have

$$A'_\pi{}^i = \begin{bmatrix} A + BD_\pi C & BC_\pi \\ B_\pi C & A_\pi \end{bmatrix}^i$$

$$= \begin{bmatrix} A + BD_\pi C & BC_\pi \\ B_\pi C & A_\pi \end{bmatrix}^{i-1} \begin{bmatrix} BD_\pi C & BC_\pi \\ B_\pi C & A_\pi \end{bmatrix} + \begin{bmatrix} A + BD_\pi C & BC_\pi \\ B_\pi C & A_\pi \end{bmatrix}^{i-1} \begin{bmatrix} A & 0_{n\times s} \\ 0_{s\times n} & 0_{s\times s} \end{bmatrix}$$

$$= \begin{bmatrix} A + BD_\pi C & BC_\pi \\ B_\pi C & A_\pi \end{bmatrix}^{i-1} \begin{bmatrix} BD_\pi C & BC_\pi \\ B_\pi C & A_\pi \end{bmatrix}$$

$$+ \begin{bmatrix} A + BD_\pi C & BC_\pi \\ B_\pi C & A_\pi \end{bmatrix}^{i-2} \begin{bmatrix} BD_\pi C & BC_\pi \\ B_\pi C & A_\pi \end{bmatrix} \begin{bmatrix} A & 0_{n\times s} \\ 0_{s\times n} & 0_{s\times s} \end{bmatrix}$$

$$+ \begin{bmatrix} A + BD_\pi C & BC_\pi \\ B_\pi C & A_\pi \end{bmatrix}^{i-2} \begin{bmatrix} A^2 & 0_{n\times s} \\ 0_{s\times n} & 0_{s\times s} \end{bmatrix}$$

$$\vdots$$

$$= \begin{bmatrix} A^i & 0_{n\times s} \\ 0_{s\times n} & 0_{s\times s} \end{bmatrix} + \sum_{j=1}^{i} A'_\pi{}^{j-1} \begin{bmatrix} BD_\pi C & BC_\pi \\ B_\pi C & A_\pi \end{bmatrix} \begin{bmatrix} A & 0_{n\times s} \\ 0_{s\times n} & 0_{s\times s} \end{bmatrix}^{i-j}$$

We use this decomposition to relate $u_t^\pi$ and $u_t^{\mathbf{M}}$. Now considering $A'_\pi{}^{i-1} B'_{\pi,w}$, for $i - 1 > 0$ we have

$$A'_\pi{}^{i-1} B'_{\pi,w} = \begin{bmatrix} A^{i-1} \\ 0_{s\times n} \end{bmatrix} + \sum_{j=1}^{i-1} A'_\pi{}^{j-1} \begin{bmatrix} BD_\pi C & BC_\pi \\ B_\pi C & A_\pi \end{bmatrix} \begin{bmatrix} A^{i-1-j} \\ 0_{s\times n} \end{bmatrix} = \begin{bmatrix} A^{i-1} \\ 0_{s\times n} \end{bmatrix} + \sum_{j=1}^{i-1} A'_\pi{}^{j-1} B'_{\pi,z} C A^{i-1-j}$$

Using this equality in the derivation of $u_t^\pi$ we derive,

$$u_t^\pi = D_\pi z_t + \sum_{i=1}^{t-1} C'_{\pi,u} A'_\pi{}^{i-1} B'_{\pi,z} z_{t-i} + D_\pi C w_{t-1}$$

$$+ \sum_{i=2}^{t-1} \begin{bmatrix} D_\pi C & C_\pi \end{bmatrix} \begin{bmatrix} A^{i-1} \\ 0_{s\times n} \end{bmatrix} w_{t-i} + \sum_{i=2}^{t-1} C'_{\pi,u} \left( \sum_{j=1}^{i-1} A'_\pi{}^{j-1} B'_{\pi,z} C A^{i-1-j} \right) w_{t-i}$$

$$= D_\pi z_t + \sum_{i=1}^{t-1} C'_{\pi,u} A'_\pi{}^{i-1} B'_{\pi,z} z_{t-i} + \sum_{i=1}^{t-1} D_\pi C A^{i-1} w_{t-i} + \sum_{i=2}^{t-1} \sum_{j=1}^{i-1} C'_{\pi,u} A'_\pi{}^{j-1} B'_{\pi,z} C A^{i-1-j} w_{t-i}$$

Note that $b_t(\mathbf{G}) = z_t + \sum_{i=1}^{t-1} C A^{t-i-1} w_i = z_t + \sum_{i=1}^{t-1} C A^{i-1} w_{t-i}$. Inspired by this expression, we rearrange the previous sum as follows:

$$u_t^\pi = D_\pi \left( z_t + \sum_{i=1}^{t-1} C A^{i-1} w_{t-i} \right) + \sum_{i=1}^{t-1} C'_{\pi,u} A'_\pi{}^{i-1} B'_{\pi,z} z_{t-i} + \sum_{i=2}^{t-1} \sum_{j=1}^{i-1} C'_{\pi,u} A'_\pi{}^{j-1} B'_{\pi,z} C A^{i-1-j} w_{t-i}$$

$$= D_\pi \left( z_t + \sum_{i=1}^{t-1} C A^{i-1} w_{t-i} \right) + \sum_{i=1}^{t-1} C'_{\pi,u} A'_\pi{}^{i-1} B'_{\pi,z} z_{t-i} + \sum_{j=1}^{t-2} \sum_{i=j+1}^{t-1} C'_{\pi,u} A'_\pi{}^{j-1} B'_{\pi,z} C A^{i-1-j} w_{t-i}$$

$$= D_\pi \left( z_t + \sum_{i=1}^{t-1} C A^{i-1} w_{t-i} \right) + \sum_{i=1}^{t-1} C'_{\pi,u} A'_\pi{}^{i-1} B'_{\pi,z} z_{t-i} + \sum_{j=1}^{t-2} C'_{\pi,u} A'_\pi{}^{j-1} B'_{\pi,z} \sum_{i=1}^{t-j-1} C A^{t-j-i-1} w_i$$

$$= D_\pi b_t + \sum_{i=1}^{t-1} C'_{\pi,u} A'_\pi{}^{i-1} B'_{\pi,z} b_{t-i}$$

Now setting $M^{[0]} = D_\pi$, and $M^{[i]} = C'_{\pi,u} {A'_\pi}^{i-1} B'_{\pi,z}$ for all $0 < i < H'$, we conclude that for any LDC policy $\pi \in \Pi$, there exists at least one length $H'$ DFC policy $\mathbf{M}(H')$ such that

$$u_t^\pi - u_t^\mathbf{M} = \sum_{i=H'}^{t} C'_{\pi,u} {A'_\pi}^{i-1} B'_{\pi,z} b_{t-i}$$

Using Cauchy Schwarz inequality we have

$$\|u_t^\pi - u_t^\mathbf{M}\| \leq \left\| \sum_{i=H'}^{t} C'_{\pi,u} {A'_\pi}^{i-1} B'_{\pi,z} b_{t-i} \right\| \leq \psi(H') \kappa_b$$

which states the first half of the Lemma.

Using the definition of $y_t^\pi$ in (21), we have

$$y_t^\pi = z_t + \sum_{i=1}^{t-1} C A^{t-i-1} w_i + \sum_{i}^{t-1} G^{[i]} u_{t-i}^\pi.$$

Similarly for $y_t^\mathbf{M}$ we have,

$$y_t^\mathbf{M} = z_t + \sum_{i=1}^{t-1} C A^{t-i-1} w_i + \sum_{i}^{t-1} G^{[i]} u_{t-i}^\mathbf{M}.$$

Subtracting these two equations, we derive,

$$y_t^\pi - y_t^\mathbf{M} = \sum_{i}^{t-1} G^{[i]} u_{t-i}^\pi - \sum_{i}^{t-1} G^{[i]} u_{t-i}^\mathbf{M} = \sum_{i}^{t-1} G^{[i]} (u_{t-i}^\pi - u_{t-i}^\mathbf{M})$$

resulting in

$$\|y_t^\pi - y_t^\mathbf{M}\| \leq \psi(H') \kappa_\mathbf{G} \kappa_b$$

which states the second half of the Lemma. $\qquad \square$

This lemma further entails that any stabilizing LDC can be well approximated by a DFC that belongs to the following set of DFCs,

$$\mathcal{M}(H', \kappa_\psi) = \left\{ \mathbf{M}(H') := \{M^{[i]}\}_{i=0}^{H'-1} : \sum_{i \geq 0}^{H'-1} \|M^{[i]}\| \leq \kappa_\psi \right\},$$

indicating that using the class of DFC policies as an approximation to LDC policies is justified.

## B  Model Parameters Identification Procedure, SYSID

Algorithm 1 gives the model parameters identification algorithm, SYSID, that is executed after recovering $\widehat{\mathcal{G}}_{\mathbf{y},\mathbf{i}}$ in the beginning of each epoch $i$. SYSID is similar to Ho-Kalman method [16] and estimates the system parameters from $\widehat{\mathcal{G}}_{\mathbf{y},\mathbf{i}}$.

First of all, notice that $\mathcal{G}_\mathbf{y} = [\mathbf{F}, \mathbf{G}]$ where

$$\mathbf{F} = \begin{bmatrix} CF, & C\bar{A}F, & \ldots, & C\bar{A}^{H_e-1}F \end{bmatrix} \in \mathbb{R}^{m \times mH_e},$$
$$\mathbf{G} = \begin{bmatrix} CB, & C\bar{A}B, & \ldots, & C\bar{A}^{H_e-1}B \end{bmatrix} \in \mathbb{R}^{m \times pH_e}.$$

Given the estimate for the truncated ARX model

$$\widehat{\mathcal{G}}_{\mathbf{y},\mathbf{i}} = [\hat{\mathbf{F}}_{\mathbf{i},\mathbf{1}}, \ldots, \hat{\mathbf{F}}_{\mathbf{i},\mathbf{H_e}}, \hat{\mathbf{G}}_{\mathbf{i},\mathbf{1}}, \ldots, \hat{\mathbf{G}}_{\mathbf{i},\mathbf{H_e}}],$$

where $\hat{\mathbf{F}}_{\mathbf{i},\mathbf{j}}$ is the $j$'th $m \times m$ block of $\hat{\mathbf{F}}_\mathbf{i}$, and $\hat{\mathbf{G}}_{\mathbf{i},\mathbf{j}}$ is the $j$'th $m \times p$ block of $\hat{\mathbf{G}}_\mathbf{i}$ for all $1 \leq j \leq H_e$. SYSID constructs two $d_1 \times (d_2 + 1)$ Hankel matrices $\mathcal{H}_{\hat{\mathbf{F}}_\mathbf{i}}$ and $\mathcal{H}_{\hat{\mathbf{G}}_\mathbf{i}}$ such that $(j, k)$th block of Hankel matrix is $\hat{\mathbf{F}}_{\mathbf{i},(\mathbf{j}+\mathbf{k}-\mathbf{1})}$ and $\hat{\mathbf{G}}_{\mathbf{i},(\mathbf{j}+\mathbf{k}-\mathbf{1})}$ respectively. Then, it forms the following matrix $\hat{\mathcal{H}}_i$.

$$\hat{\mathcal{H}}_i = \begin{bmatrix} \mathcal{H}_{\hat{\mathbf{F}}_\mathbf{i}}, & \mathcal{H}_{\hat{\mathbf{G}}_\mathbf{i}} \end{bmatrix}.$$

Recall that from $(A, F)$-controllability of $\Theta$, we have that $\|T\bar{A}T^{-1}\| \le v < 1$ for some similarity transformation $T$ and the dimension of latent state, $n$, is the order of the system for the observable and controllable system. For $H_e \ge \max\left\{2n+1, \frac{\log(c_H T^2 \sqrt{m}/\sqrt{\lambda})}{\log(1/v)}\right\}$ for some problem dependent constant $c_H$, we can pick $d_1 \ge n$ and $d_2 \ge n$ such $d_1 + d_2 + 1 = H_e$. This guarantees that the system identification problem is well-conditioned. Using Definition 2.1, if the input to the SYSID was $\mathcal{G}_{\mathbf{y}} = [\mathbf{F}, \mathbf{G}]$ then constructed Hankel matrix, $\mathcal{H}$ would be rank $n$,

$$\mathcal{H} = [C^\top, \ \ldots, \ (C\bar{A}^{d_1-1})^\top]^\top [F, \ \ldots, \ \bar{A}^{d_2}F, \ B, \ \ldots, \ \bar{A}^{d_2}B]$$
$$= \mathbf{O}(\bar{A}, C, d_1) \ [\mathbf{C}(\bar{A}, F, d_2+1), \quad \bar{A}^{d_2}F, \quad \mathbf{C}(\bar{A}, B, d_2+1), \quad \bar{A}^{d_2}B]$$
$$= \mathbf{O}(\bar{A}, C, d_1) \ [F, \quad \bar{A}\mathbf{C}(\bar{A}, F, d_2+1), \quad B, \quad \bar{A}\mathbf{C}(\bar{A}, B, d_2+1)].$$

where for all $k \ge n$, $\mathbf{C}(A, B, k)$ defines the extended $(A, B)$ controllability matrix and $\mathbf{O}(A, C, k)$ defines the extended $(A, C)$ observability matrix. Notice that $\mathcal{G}_{\mathbf{y}}$ and $\mathcal{H}$ are uniquely identifiable for a given system $\Theta$, whereas for any invertible $\mathbf{T} \in \mathbb{R}^{n \times n}$, the system resulting from

$$A' = \mathbf{T}^{-1}A\mathbf{T}, \ B' = \mathbf{T}^{-1}B, \ C' = C\mathbf{T}, \ F' = \mathbf{T}^{-1}F$$

gives the same $\mathcal{G}_{\mathbf{y}}$ and $\mathcal{H}$. Similar to Ho-Kalman algorithm, SYSID computes the SVD of $\widehat{\mathcal{G}}_{\mathbf{y},\mathbf{i}}$ and estimates the extended observability and controllability matrices and eventually system parameters up to similarity transformation. To this end, SYSID constructs $\hat{\mathcal{H}}_i^-$ by discarding $(d_2+1)$th and $(2d_2+2)$th block columns of $\hat{\mathcal{H}}_i$, $i.e.$ if it was $\mathcal{H}$ then we have,

$$\mathcal{H}^- = \mathbf{O}(\bar{A}, C, d_1) \ [\mathbf{C}(\bar{A}, F, d_2+1), \quad \mathbf{C}(\bar{A}, B, d_2+1)].$$

The SYSID procedure then calculates $\hat{\mathcal{N}}_i$, the best rank-$n$ approximation of $\hat{\mathcal{H}}_i^-$, obtained by setting its all but top $n$ singular values to zero. The estimates of $\mathbf{O}(\bar{A}, C, d_1)$, $\mathbf{C}(\bar{A}, F, d_2+1)$ and $\mathbf{C}(\bar{A}, B, d_2+1)$ are given as

$$\hat{\mathcal{N}}_i = \mathbf{U}_{\mathbf{i}}\boldsymbol{\Sigma}_{\mathbf{i}}^{1/2} \, \boldsymbol{\Sigma}_{\mathbf{i}}^{1/2}\mathbf{V}_{\mathbf{i}}^\top = \hat{\mathbf{O}}_{\mathbf{i}}(\bar{A}, C, d_1) \ [\hat{\mathbf{C}}_{\mathbf{i}}(\bar{A}, F, d_2+1), \quad \hat{\mathbf{C}}_{\mathbf{i}}(\bar{A}, B, d_2+1)].$$

From these estimates SYSID recovers $\hat{C}_i$ as the first $m \times n$ block of $\hat{\mathbf{O}}_{\mathbf{i}}(\bar{A}, C, d_1)$, $\hat{B}_i$ as the first $n \times p$ block of $\hat{\mathbf{C}}_{\mathbf{i}}(\bar{A}, B, d_2+1)$ and $\hat{F}_i$ as the first $n \times m$ block of $\hat{\mathbf{C}}_{\mathbf{i}}(\bar{A}, F, d_2+1)$. Let $\hat{\mathcal{H}}_i^+$ be the matrix obtained by discarding 1st and $(d_2+2)$th block columns of $\hat{\mathcal{H}}_i$, $i.e.$ if it was $\mathcal{H}$ then

$$\mathcal{H}^+ = \mathbf{O}(\bar{A}, C, d_1) \ \bar{A} \ [\mathbf{C}(\bar{A}, F, d_2+1), \quad \mathbf{C}(\bar{A}, B, d_2+1)].$$

Therefore, SYSID recovers $\hat{\bar{A}}_i$ as,

$$\hat{\bar{A}}_i = \hat{\mathbf{O}}_{\mathbf{i}}^\dagger(\bar{A}, C, d_1) \ \hat{\mathcal{H}}_t^+ \ [\hat{\mathbf{C}}_{\mathbf{t}}(\bar{A}, F, d_2+1), \quad \hat{\mathbf{C}}_{\mathbf{t}}(\bar{A}, B, d_2+1)]^\dagger.$$

Using the definition of $\bar{A} = A - FC$, the algorithm obtains $\hat{A}_t = \hat{\bar{A}}_t + \hat{F}_t\hat{C}_t$.

---

**Algorithm 1** SYSID

---

1: **Input:** $\widehat{\mathcal{G}}_{\mathbf{y},\mathbf{i}}$, $H_e$, system order $n$, $d_1, d_2$ such that $d_1 + d_2 + 1 = H_e$
2: Form two $d_1 \times (d_2+1)$ Hankel matrices $\mathcal{H}_{\hat{\mathbf{F}}_{\mathbf{i}}}$ and $\mathcal{H}_{\hat{\mathbf{G}}_{\mathbf{i}}}$ from $\widehat{\mathcal{G}}_{\mathbf{y},\mathbf{i}} = [\hat{\mathbf{F}}_{\mathbf{i},\mathbf{1}}, \ldots, \hat{\mathbf{F}}_{\mathbf{i},\mathbf{H_e}}, \hat{\mathbf{G}}_{\mathbf{i},\mathbf{1}}, \ldots, \hat{\mathbf{G}}_{\mathbf{i},\mathbf{H_e}}]$, and construct $\hat{\mathcal{H}}_i = \left[\mathcal{H}_{\hat{\mathbf{F}}_{\mathbf{i}}}, \ \mathcal{H}_{\hat{\mathbf{G}}_{\mathbf{i}}}\right] \in \mathbb{R}^{md_1 \times (m+p)(d_2+1)}$
3: Obtain $\hat{\mathcal{H}}_i^-$ by discarding $(d_2+1)$th and $(2d_2+2)$th block columns of $\hat{\mathcal{H}}_i$
4: Using SVD obtain $\hat{\mathcal{N}}_i \in \mathbb{R}^{md_1 \times (m+p)d_2}$, the best rank-$n$ approximation of $\hat{\mathcal{H}}_i^-$
5: Obtain $\mathbf{U}_{\mathbf{i}}, \boldsymbol{\Sigma}_{\mathbf{i}}, \mathbf{V}_{\mathbf{i}} = \text{SVD}(\hat{\mathcal{N}}_i)$
6: Construct $\hat{\mathbf{O}}_{\mathbf{i}}(\bar{A}, C, d_1) = \mathbf{U}_{\mathbf{i}}\boldsymbol{\Sigma}_{\mathbf{t}}^{1/2} \in \mathbb{R}^{md_1 \times n}$
7: Construct $[\hat{\mathbf{C}}_{\mathbf{i}}(\bar{A}, F, d_2+1), \quad \hat{\mathbf{C}}_{\mathbf{i}}(\bar{A}, B, d_2+1)] = \boldsymbol{\Sigma}_{\mathbf{i}}^{1/2}\mathbf{V}_{\mathbf{i}} \in \mathbb{R}^{n \times (m+p)d_2}$
8: Obtain $\hat{C}_i \in \mathbb{R}^{m \times n}$, the first $m$ rows of $\hat{\mathbf{O}}_{\mathbf{i}}(\bar{A}, C, d_1)$
9: Obtain $\hat{B}_i \in \mathbb{R}^{n \times p}$, the first $p$ columns of $\hat{\mathbf{C}}_{\mathbf{i}}(\bar{A}, B, d_2+1)$
10: Obtain $\hat{F}_i \in \mathbb{R}^{n \times m}$, the first $m$ columns of $\hat{\mathbf{C}}_{\mathbf{i}}(\bar{A}, F, d_2+1)$
11: Obtain $\hat{\mathcal{H}}_i^+$ by discarding 1st and $(d_2+2)$th block columns of $\hat{\mathcal{H}}_i$
12: Obtain $\hat{\bar{A}}_i = \hat{\mathbf{O}}_{\mathbf{i}}^\dagger(\bar{A}, C, d_1) \ \hat{\mathcal{H}}_i^+ \ [\hat{\mathbf{C}}_{\mathbf{i}}(\bar{A}, F, d_2+1), \quad \hat{\mathbf{C}}_{\mathbf{i}}(\bar{A}, B, d_2+1)]^\dagger$
13: Obtain $\hat{A}_i = \hat{\bar{A}}_i + \hat{F}_i\hat{C}_i$

---

## C  ADAPTON

---

**Algorithm 2** ADAPTON

---

1: **Input:** $T$, $H$, $H'$, $T_w$, $\mathcal{M}$
2: —— WARM-UP ————————————————————————————————
3: **for** $t = 1, \dots, T_w$ **do**
4:     Deploy $u_t \sim \mathcal{N}(0, \sigma_u^2 I)$
5: Store $\mathcal{D}_{T_w} = \{y_t, u_t\}_{t=1}^{T_w}$, set $t_1 = T_{base} = T_w$, $t = T_{base} + 1$, and $\mathbf{M}_t$ as any member of $\mathcal{M}$
6: —— ADAPTIVE CONTROL ————————————————————————
7: **for** $i = 1, 2, \dots$ **do**
8:     Solve (10) using $\mathcal{D}_t$, estimate $\hat{A}_i$, $\hat{B}_i$, $\hat{C}_i$ using SYSID (Alg. 1) and construct $\widehat{\mathbf{G}}_i(H)$
9:     Compute $b_\tau(\widehat{\mathbf{G}}_i) := y_\tau - \sum_{j=1}^{H} \widehat{G}_i^{[j]} u_{\tau-j}, \forall \tau \leq t$
10:    **while** $t \leq t_i + 2^{i-1} T_{base}$ *and* $t \leq T$ **do**
11:        Observe $y_t$, and compute $b_t(\widehat{\mathbf{G}}_i) := y_t - \sum_{j=1}^{H} \widehat{G}_i^{[j]} u_{t-j}$
12:        Commit to $u_t = \sum_{j=0}^{H'-1} M_t^{[j]} b_{t-j}(\widehat{\mathbf{G}}_i)$, observe $\ell_t$, and pay a cost of $\ell_t(y_t, u_t)$
13:        Update $\mathbf{M}_{t+1} = proj_{\mathcal{M}}\left(\mathbf{M}_t - \eta_t \nabla f_t\left(\mathbf{M}_t, \widehat{\mathbf{G}}_i\right)\right)$, $\mathcal{D}_{t+1} = \mathcal{D}_t \cup \{y_t, u_t\}$, set $t = t+1$
14:    $t_{i+1} = t_i + 2^{i-1} T_{base}$

---

## D  Proof of Theorem 1

In this section, we provide the proof of Theorem 1 with precise expressions. In Appendix D.1, we show the self-normalized error bound on the (10), Theorem 3. In Appendix D.2, assuming persistence of excitation, we convert the self-normalized bound into a Frobenius norm bound to be used for parameter estimation error bounds in Appendix D.3, Theorem 4. Finally, we consider the Markov parameter estimates constructed via model parameter estimates in Appendix D.4, which concludes the proof of Theorem 1.

### D.1  Self-Normalized Bound on Finite Sample Estimation Error of (10)

First consider the effect of truncation bias term, $C\bar{A}^{H_e} x_{t-H_e}$. Notice that $\bar{A}$ is stable due to $(A, F)$-controllability of $\Theta$ [17], *i.e.,* $\|T\bar{A}T^{-1}\| \leq \upsilon < 1$ for some similarity transformation $T$. Thus, $C\bar{A}^{H_e} x_{t-H_e}$ is order of $\upsilon^H$. In order to get consistent estimation, for some problem dependent constant $c_H$, we set $H_e \geq \frac{\log(c_H T^2 \sqrt{m}/\sqrt{\lambda})}{\log(1/\upsilon)}$, resulting in a negligible bias term of order $1/T^2$. Using this we first obtain a self-normalized finite sample estimation error of (10):

**Theorem 3** (Self-normalized Estimation Error). *Let $\widehat{\mathcal{G}}_\mathbf{y}$ be the solution to (10) at time $\tau$. For the given choice of $H_e$, define*

$$V_\tau = \lambda I + \sum_{i=H_e}^{\tau} \phi_i \phi_i^\top.$$

*Let $\|\mathbf{M}\|_F \leq S$. For $\delta \in (0, 1)$, with probability at least $1 - \delta$, for all $t \leq \tau$, $\mathcal{G}_\mathbf{y}$ lies in the set $\mathcal{C}_{\mathcal{G}_\mathbf{y}}(t)$, where*

$$\mathcal{C}_{\mathcal{G}_\mathbf{y}}(t) = \{\mathcal{G}_\mathbf{y}' : \mathrm{Tr}((\widehat{\mathcal{G}}_\mathbf{y} - \mathcal{G}_\mathbf{y}') V_t (\widehat{\mathcal{G}}_\mathbf{y} - \mathcal{G}_\mathbf{y}')^\top) \leq \beta_\tau\},$$

*for $\beta_\tau$ defined as follows,*

$$\beta_\tau = \left(\sqrt{m\|C\Sigma C^\top + \sigma_z^2 I\| \log\left(\frac{\det(V_\tau)^{1/2}}{\delta \det(\lambda I)^{1/2}}\right)} + S\sqrt{\lambda} + \frac{\tau\sqrt{H_e}}{T^2}\right)^2.$$

*Proof.* For a single input-output trajectory $\{y_t, u_t\}_{t=1}^\tau$, where $\tau \leq T$, using the representation in (8), we can write the following for the given system,

$$Y_\tau = \Phi_\tau \mathcal{G}_\mathbf{y}^\top + \underbrace{E_\tau + N_\tau}_{\text{Noise}} \qquad \text{where} \tag{22}$$

$$\mathcal{G}_{\mathbf{y}} = \begin{bmatrix} CF, & C\bar{A}F, & \ldots, & C\bar{A}^{H_e-1}F, & CB, & C\bar{A}B, & \ldots, & C\bar{A}^{H_e-1}B \end{bmatrix} \in \mathbb{R}^{m\times(m+p)H_e}$$

$$Y_\tau = [y_{H_e},\ y_{H_e+1},\ \ldots,\ y_\tau]^\top \in \mathbb{R}^{(\tau-H_e)\times m}$$

$$\Phi_\tau = [\phi_{H_e},\ \phi_{H_e+1},\ \ldots,\ \phi_\tau]^\top \in \mathbb{R}^{(\tau-H_e)\times(m+p)H_e}$$

$$E_\tau = [e_{H_e},\ e_{H_e+1},\ \ldots,\ e_\tau]^\top \in \mathbb{R}^{(\tau-H_e)\times m}$$

$$N_\tau = \begin{bmatrix} C\bar{A}^{H_e}x_0,\ C\bar{A}^{H_e}x_1,\ldots,C\bar{A}^{H_e}x_{\tau-H_e} \end{bmatrix}^\top \in \mathbb{R}^{(\tau-H_e)\times m}.$$

$\widehat{\mathcal{G}}_{\mathbf{y}}$ is the solution to (10), *i.e.,* $\min_X \|Y_\tau - \Phi_\tau X^\top\|_F^2 + \lambda\|X\|_F^2$. Hence, we get $\widehat{\mathcal{G}}_{\mathbf{y}}^\top = (\Phi_\tau^\top\Phi_\tau + \lambda I)^{-1}\Phi_\tau^\top Y_\tau$.

$$
\begin{aligned}
\widehat{\mathcal{G}}_{\mathbf{y}} &= \left[ (\Phi_\tau^\top\Phi_\tau + \lambda I)^{-1}\Phi_\tau^\top(\Phi_\tau\mathcal{G}_{\mathbf{y}}^\top + E_\tau + N_\tau) \right]^\top \\
&= \left[ (\Phi_\tau^\top\Phi_\tau + \lambda I)^{-1}\Phi_\tau^\top(E_\tau + N_\tau) + (\Phi_\tau^\top\Phi_\tau + \lambda I)^{-1}\Phi_\tau^\top\Phi_\tau\mathcal{G}_{\mathbf{y}}^\top \right. \\
&\qquad \left. + \lambda(\Phi_\tau^\top\Phi_\tau + \lambda I)^{-1}\mathcal{G}_{\mathbf{y}}^\top - \lambda(\Phi_\tau^\top\Phi_\tau + \lambda I)^{-1}\mathcal{G}_{\mathbf{y}}^\top \right]^\top \\
&= \left[ (\Phi_\tau^\top\Phi_\tau + \lambda I)^{-1}\Phi_\tau^\top E_\tau + (\Phi_\tau^\top\Phi_\tau + \lambda I)^{-1}\Phi_\tau^\top N_\tau + \mathcal{G}_{\mathbf{y}}^\top - \lambda(\Phi_\tau^\top\Phi_\tau + \lambda I)^{-1}\mathcal{G}_{\mathbf{y}}^\top \right]^\top
\end{aligned}
$$

Using $\widehat{\mathcal{G}}_{\mathbf{y}}$, we get

$$
\begin{aligned}
&|\operatorname{Tr}(X(\widehat{\mathcal{G}}_{\mathbf{y}} - \mathcal{G}_{\mathbf{y}})^\top)| &&(23)\\
&= |\operatorname{Tr}(X(\Phi_\tau^\top\Phi_\tau + \lambda I)^{-1}\Phi_\tau^\top E_\tau) + \operatorname{Tr}(X(\Phi_\tau^\top\Phi_\tau + \lambda I)^{-1}\Phi_\tau^\top N_\tau) - \lambda\operatorname{Tr}(X(\Phi_\tau^\top\Phi_\tau + \lambda I)^{-1}\mathcal{G}_{\mathbf{y}}^\top)| \\
&\le |\operatorname{Tr}(X(\Phi_\tau^\top\Phi_\tau + \lambda I)^{-1}\Phi_\tau^\top E_\tau)| + |\operatorname{Tr}(X(\Phi_\tau^\top\Phi_\tau + \lambda I)^{-1}\Phi_\tau^\top N_\tau)| + \lambda|\operatorname{Tr}(X(\Phi_\tau^\top\Phi_\tau + \lambda I)^{-1}\mathcal{G}_{\mathbf{y}}^\top)| \\
&\le \sqrt{\operatorname{Tr}(X(\Phi_\tau^\top\Phi_\tau + \lambda I)^{-1}X^\top)\operatorname{Tr}(E_\tau^\top\Phi_\tau(\Phi_\tau^\top\Phi_\tau + \lambda I)^{-1}\Phi_\tau^\top E_\tau)} &&(24)\\
&\quad + \sqrt{\operatorname{Tr}(X(\Phi_\tau^\top\Phi_\tau + \lambda I)^{-1}X^\top)\operatorname{Tr}(N_\tau^\top\Phi_\tau(\Phi_\tau^\top\Phi_\tau + \lambda I)^{-1}\Phi_\tau^\top N_\tau)} \\
&\quad + \lambda\sqrt{\operatorname{Tr}(X(\Phi_\tau^\top\Phi_\tau + \lambda I)^{-1}X^\top)\operatorname{Tr}(\mathcal{G}_{\mathbf{y}}(\Phi_\tau^\top\Phi_\tau + \lambda I)^{-1}\mathcal{G}_{\mathbf{y}}^\top)} \\
&= \sqrt{\operatorname{Tr}(X(\Phi_\tau^\top\Phi_\tau + \lambda I)^{-1}X^\top)} \times \\
&\quad \left[\sqrt{\operatorname{Tr}(E_\tau^\top\Phi_\tau(\Phi_\tau^\top\Phi_\tau + \lambda I)^{-1}\Phi_\tau^\top E_\tau)} + \sqrt{\operatorname{Tr}(N_\tau^\top\Phi_\tau(\Phi_\tau^\top\Phi_\tau + \lambda I)^{-1}\Phi_\tau^\top N_\tau)} + \lambda\sqrt{\operatorname{Tr}(\mathcal{G}_{\mathbf{y}}(\Phi_\tau^\top\Phi_\tau + \lambda I)^{-1}\mathcal{G}_{\mathbf{y}}^\top)}\right]
\end{aligned}
$$

where (24) follows from $|\operatorname{Tr}(ABC^\top)| \le \sqrt{\operatorname{Tr}(ABA^\top)\operatorname{Tr}(CBC^\top)}$ for positive definite B due to Cauchy Schwarz (weighted inner-product). For $X = (\widehat{\mathcal{G}}_{\mathbf{y}} - \mathcal{G}_{\mathbf{y}})(\Phi_\tau^\top\Phi_\tau + \lambda I)$, we get

$$\sqrt{\operatorname{Tr}((\widehat{\mathcal{G}}_{\mathbf{y}} - \mathcal{G}_{\mathbf{y}})V_\tau(\widehat{\mathcal{G}}_{\mathbf{y}} - \mathcal{G}_{\mathbf{y}})^\top)} \le \sqrt{\operatorname{Tr}(E_\tau^\top\Phi_\tau V_\tau^{-1}\Phi_\tau^\top E_\tau)} + \sqrt{\operatorname{Tr}(N_\tau^\top\Phi_\tau V_\tau^{-1}\Phi_\tau^\top N_\tau)} + \sqrt{\lambda}\|\mathcal{G}_{\mathbf{y}}\|_F \tag{25}$$

where $V_\tau$ is the regularized design matrix at time $\tau$. Let $\max_{i\le\tau}\|\phi_i\| \le \Upsilon\sqrt{H_e}$ and $\max_{H_e\le i\le\tau}\|x_i\| \le \mathcal{X}$, *i.e.,* in data collection bounded inputs are used. The first term on the right hand side of (25) can be bounded using Theorem 7 since $e_t$ is $\|C\Sigma C^\top + \sigma_z^2 I\|$-sub-Gaussian vector. Therefore, for $\delta \in (0,1)$, with probability at least $1 - \delta$,

$$\sqrt{\operatorname{Tr}(E_\tau^\top\Phi_t V_\tau^{-1}\Phi_\tau^\top E_\tau)} \le \sqrt{m\|C\Sigma C^\top + \sigma_z^2 I\|\log\left(\frac{\det(V_\tau)^{1/2}}{\delta\det(\lambda I)^{1/2}}\right)} \tag{26}$$

For the second term,

$$\sqrt{\mathrm{Tr}(N_\tau^\top \Phi_\tau V_\tau^{-1} \Phi_\tau^\top N_\tau)} \le \frac{1}{\sqrt{\lambda}} \|N_\tau^\top \Phi_\tau\|_F \le \sqrt{\frac{m}{\lambda}} \left\| \sum_{i=H_e}^{\tau} \phi_i (C\bar{A}^{H_e} x_{i-H_e})^\top \right\|$$

$$\le \tau \sqrt{\frac{m}{\lambda}} \max_{i \le \tau} \left\| \phi_i (C\bar{A}^{H_e} x_{i-H_e})^\top \right\|$$

$$\le \tau \sqrt{\frac{m}{\lambda}} \|C\| \upsilon^{H_e} \max_{i \le \tau} \|\phi_i\| \|x_{i-H_e}\|$$

$$\le \tau \sqrt{\frac{m}{\lambda}} \|C\| \upsilon^{H_e} \Upsilon \sqrt{H_e} \mathcal{X}.$$

Picking $H_e = \frac{2\log(T) + \log(\Upsilon \mathcal{X}) + 0.5\log(m/\lambda) + \log(\|C\|)}{\log(1/\upsilon)}$ gives

$$\sqrt{\mathrm{Tr}(N_\tau^\top \Phi_\tau V_\tau^{-1} \Phi_\tau^\top N_\tau)} \le \frac{\tau}{T^2} \sqrt{H_e}. \tag{27}$$

Combining (26) and (27) gives the self-normalized estimation error bound state in the theorem. $\square$

### D.2 Frobenius Norm Bound on Finite Sample Estimation Error of (10)

Using this result, we have

$$\sigma_{\min}(V_\tau) \|\widehat{\mathcal{G}}_\mathbf{y} - \mathcal{G}_\mathbf{y}\|_F^2 \le \mathrm{Tr}((\widehat{\mathcal{G}}_\mathbf{y} - \mathcal{G}_\mathbf{y}) V_t (\widehat{\mathcal{G}}_\mathbf{y} - \mathcal{G}_\mathbf{y})^\top)$$

$$\le \left( \sqrt{m\|C\Sigma C^\top + \sigma_z^2 I\| \log\left( \frac{\det(V_\tau)^{1/2}}{\delta \det(\lambda I)^{1/2}} \right)} + S\sqrt{\lambda} + \frac{\tau\sqrt{H_e}}{T^2} \right)^2$$

For persistently exciting inputs, *i.e.*, $\sigma_{\min}(V_\tau) \ge \sigma_\star^2 \tau$ for $\sigma_\star > 0$, using the boundedness of $\phi_i$, *i.e.*, $\max_{i \le \tau} \|\phi_i\| \le \Upsilon \sqrt{H_e}$, we get,

$$\|\widehat{\mathcal{G}}_\mathbf{y} - \mathcal{G}_\mathbf{y}\|_F \le \frac{\sqrt{m\|C\Sigma C^\top + \sigma_z^2 I\| \left( \log(1/\delta) + \frac{H_e(m+p)}{2} \log\left( \frac{\lambda(m+p) + \tau \Upsilon^2}{\lambda(m+p)} \right) \right)} + S\sqrt{\lambda} + \frac{\sqrt{H_e}}{T}}{\sigma_\star \sqrt{\tau}}$$

This result shows that under persistent of excitation, the novel least squares problem provides consistent estimates and the estimation error is $\tilde{\mathcal{O}}(1/\sqrt{T})$ after $T$ samples.

### D.3 Bound on Model Parameters Estimation Errors

After estimating $\widehat{\mathcal{G}}_\mathbf{y}$, we deploy SYSID (Appendix B) to recover the unknown system parameters. The outline of the algorithm is given in Algorithm 1. Note that the system is order $n$ and minimal in the sense that the system cannot be described by a state-space model of order less than $n$. Define $T_{\mathcal{G}_\mathbf{y}}$ such that at $T_{\mathcal{G}_\mathbf{y}}$, $\|\widehat{\mathcal{G}}_\mathbf{y} - \mathcal{G}_\mathbf{y}\| \le 1$. Let $T_N = T_{\mathcal{G}_\mathbf{y}} \frac{8H_e}{\sigma_n^2(\mathcal{H})}$, $T_B = T_{\mathcal{G}_\mathbf{y}} \frac{20nH_e}{\sigma_n(\mathcal{H})}$. We have the following result on the model parameter estimates:

**Theorem 4** (Model Parameters Estimation Error). *Let $\mathcal{H}$ be the concatenation of two Hankel matrices obtained from $\mathcal{G}_\mathbf{y}$. Let $\bar{A}, \bar{B}, \bar{C}, \bar{F}$ be the system parameters that SYSID provides for $\mathcal{G}_\mathbf{y}$. At time step $t$, let $\hat{A}_t, \hat{B}_t, \hat{C}_t, \hat{F}_t$ denote the system parameters obtained by SYSID using $\widehat{\mathcal{G}}_\mathbf{y}$. For the described system in the main text $\mathcal{H}$ is rank-$n$ i.e., due to controllability-observability assumption. For $t \ge \max\{T_{\mathcal{G}_\mathbf{y}}, T_N, T_B\}$, for the given choice of $H_e$, there exists a unitary matrix $\mathbf{T} \in \mathbb{R}^{n \times n}$ such that, $\bar{\Theta} = (\bar{A}, \bar{B}, \bar{C}, \bar{F}) \in (\mathcal{C}_A \times \mathcal{C}_B \times \mathcal{C}_C \times \mathcal{C}_F)$ where*

$$\mathcal{C}_A(t) = \left\{ A' \in \mathbb{R}^{n \times n} : \|\hat{A}_t - \mathbf{T}^\top A' \mathbf{T}\| \le \beta_A(t) \right\},$$

$$\mathcal{C}_B(t) = \left\{ B' \in \mathbb{R}^{n \times p} : \|\hat{B}_t - \mathbf{T}^\top B'\| \le \beta_B(t) \right\},$$

$$\mathcal{C}_C(t) = \left\{ C' \in \mathbb{R}^{m \times n} : \|\hat{C}_t - C' \mathbf{T}\| \le \beta_C(t) \right\},$$

$$\mathcal{C}_L(t) = \left\{ L' \in \mathbb{R}^{p \times m} : \|\hat{F}_t - \mathbf{T}^\top F'\| \le \beta_F(t) \right\},$$

*for*

$$\beta_A(t) = c_1 \left( \frac{\sqrt{nH_e}(\|\mathcal{H}\| + \sigma_n(\mathcal{H}))}{\sigma_n^2(\mathcal{H})} \right) \|\widehat{\mathcal{G}}_{\mathbf{y}} - \mathcal{G}_{\mathbf{y}}\|, \quad \beta_B(t) = \beta_C(t) = \beta_F(t) = \sqrt{\frac{20nH_e}{\sigma_n(\mathcal{H})}} \|\widehat{\mathcal{G}}_{\mathbf{y}} - \mathcal{G}_{\mathbf{y}}\|,$$

*for some problem dependent constant $c_1$.*

*Proof.* The result follows similar steps with Oymak and Ozay [3]. The following lemma is from Oymak and Ozay [3], it will be useful in proving error bounds on system parameters and we provide it for completeness.

**Lemma D.1** ([3]). *$\mathcal{H}, \hat{\mathcal{H}}_t$ and $\mathcal{N}, \hat{\mathcal{N}}_t$ satisfies the following perturbation bounds,*

$$\max \left\{ \left\| \mathcal{H}^+ - \hat{\mathcal{H}}_t^+ \right\|, \left\| \mathcal{H}^- - \hat{\mathcal{H}}_t^- \right\| \right\} \leq \|\mathcal{H} - \hat{\mathcal{H}}_t\| \leq \sqrt{\min \{d_1, d_2 + 1\}} \|\widehat{\mathcal{G}}_{\mathbf{y}} - \mathcal{G}_{\mathbf{y}}\|$$

$$\|\mathcal{N} - \hat{\mathcal{N}}_t\| \leq 2 \left\| \mathcal{H}^- - \hat{\mathcal{H}}_t^- \right\| \leq 2\sqrt{\min \{d_1, d_2\}} \|\widehat{\mathcal{G}}_{\mathbf{y}} - \mathcal{G}_{\mathbf{y}}\|$$

The following lemma is a slight modification of Lemma B.1 in [3].

**Lemma D.2** ([3]). *Suppose $\sigma_{\min}(\mathcal{N}) \geq 2\|\mathcal{N} - \hat{\mathcal{N}}\|$ where $\sigma_{\min}(\mathcal{N})$ is the smallest nonzero singular value (i.e. $n$th largest singular value) of $N$. Let rank $n$ matrices $\mathcal{N}, \hat{\mathcal{N}}$ have singular value decompositions $\mathbf{U\Sigma V}^\top$ and $\hat{\mathbf{U}}\hat{\mathbf{\Sigma}}\hat{\mathbf{V}}^\top$ There exists an $n \times n$ unitary matrix $\mathbf{T}$ so that*

$$\left\| \mathbf{U\Sigma}^{1/2} - \hat{\mathbf{U}}\hat{\mathbf{\Sigma}}^{1/2}\mathbf{T} \right\|_F^2 + \left\| \mathbf{V\Sigma}^{1/2} - \hat{\mathbf{V}}\hat{\mathbf{\Sigma}}^{1/2}\mathbf{T} \right\|_F^2 \leq \frac{5n\|\mathcal{N} - \hat{\mathcal{N}}\|^2}{\sigma_n(\mathcal{N}) - \|\mathcal{N} - \hat{\mathcal{N}}\|}$$

For brevity, we have the following notation $\mathbf{O} = \mathbf{O}(\bar{A}, C, d_1)$, $\mathbf{C_F} = \mathbf{C}(\bar{A}, F, d_2 + 1)$, $\mathbf{C_B} = \mathbf{C}(\bar{A}, B, d_2 + 1)$, $\hat{\mathbf{O}}_{\mathbf{t}} = \hat{\mathbf{O}}_{\mathbf{t}}(\bar{A}, C, d_1)$, $\hat{\mathbf{C}}_{\mathbf{F_t}} = \hat{\mathbf{C}}_{\mathbf{t}}(\bar{A}, F, d_2 + 1)$, $\hat{\mathbf{C}}_{\mathbf{B_t}} = \hat{\mathbf{C}}_{\mathbf{t}}(\bar{A}, B, d_2 + 1)$. In the definition of $T_N$, we use $\sigma_n(H)$, due to the fact that singular values of submatrices by column partitioning are interlaced, *i.e.* $\sigma_n(\mathbf{N}) = \sigma_n(\mathbf{H}^-) \geq \sigma_n(\mathbf{H})$. Directly applying Lemma D.2 with the condition that for given $t \geq T_N$, $\sigma_{\min}(\mathcal{N}) \geq 2\|\mathcal{N} - \hat{\mathcal{N}}\|$, we can guarantee that there exists a unitary transform $\mathbf{T}$ such that

$$\left\| \hat{\mathbf{O}}_{\mathbf{t}} - \mathbf{OT} \right\|_F^2 + \left\| [\hat{\mathbf{C}}_{\mathbf{F_t}} \ \hat{\mathbf{C}}_{\mathbf{B_t}}] - \mathbf{T}^\top [\mathbf{C_F} \ \mathbf{C_B}] \right\|_F^2 \leq \frac{10n\|\mathcal{N} - \hat{\mathcal{N}}_t\|^2}{\sigma_n(\mathcal{N})}. \tag{28}$$

Since $\hat{C}_t - \bar{C}\mathbf{T}$ is a submatrix of $\hat{\mathbf{O}}_{\mathbf{t}} - \mathbf{OT}$, $\hat{B}_t - \mathbf{T}^\top \bar{B}$ is a submatrix of $\hat{\mathbf{C}}_{\mathbf{B_t}} - \mathbf{T}^\top \mathbf{C_B}$ and $\hat{F}_t - \mathbf{T}^\top \bar{F}$ is a submatrix of $\hat{\mathbf{C}}_{\mathbf{F_t}} - \mathbf{T}^\top \mathbf{C_F}$, we get the same bounds for them stated in (28). Using Lemma D.1, with the choice of $d_1, d_2 \geq \frac{H_e}{2}$, we have

$$\|\mathcal{N} - \hat{\mathcal{N}}_t\| \leq \sqrt{2H_e} \|\widehat{\mathcal{G}}_{\mathbf{y}} - \mathcal{G}_{\mathbf{y}}\|.$$

This provides the advertised bounds in the theorem:

$$\|\hat{B}_t - \mathbf{T}^\top \bar{B}\|, \|\hat{C}_t - \bar{C}\mathbf{T}\|, \|\hat{F}_t - \mathbf{T}^\top \bar{F}\| \leq \frac{\sqrt{20nH_e}\|\widehat{\mathcal{G}}_{\mathbf{y}} - \mathcal{G}_{\mathbf{y}}\|}{\sqrt{\sigma_n(\mathcal{N})}}$$

Notice that for $t \geq T_B$, we have all the terms above to be bounded by 1. In order to determine the closeness of $\hat{A}_t$ and $\bar{A}$ we first consider the closeness of $\hat{\bar{A}}_t - \mathbf{T}^\top \bar{\bar{A}}\mathbf{T}$, where $\bar{\bar{A}}$ is the output obtained by SYSID for $\bar{A}$ when the input is $\mathcal{G}_{\mathbf{y}}$. Let $X = \mathbf{OT}$ and $Y = \mathbf{T}^\top [\mathbf{C_F} \ \mathbf{C_B}]$. Thus, we have

$$\|\hat{\bar{A}}_t - \mathbf{T}^\top \bar{\bar{A}}\mathbf{T}\|_F = \|\hat{\mathbf{O}}_{\mathbf{t}}^\dagger \hat{\mathcal{H}}_t^+ [\hat{\mathbf{C}}_{\mathbf{F_t}} \ \hat{\mathbf{C}}_{\mathbf{B_t}}]^\dagger - X^\dagger \mathcal{H}^+ Y^\dagger\|_F$$

$$\leq \left\| \left( \hat{\mathbf{O}}_{\mathbf{t}}^\dagger - X^\dagger \right) \hat{\mathcal{H}}_t^+ [\hat{\mathbf{C}}_{\mathbf{F_t}} \ \hat{\mathbf{C}}_{\mathbf{B_t}}]^\dagger \right\|_F + \left\| X^\dagger \left( \hat{\mathcal{H}}_t^+ - \mathcal{H}^+ \right) [\hat{\mathbf{C}}_{\mathbf{F_t}} \ \hat{\mathbf{C}}_{\mathbf{B_t}}]^\dagger \right\|_F$$

$$+ \left\| X^\dagger \mathcal{H}^+ \left( [\hat{\mathbf{C}}_{\mathbf{F_t}} \ \hat{\mathbf{C}}_{\mathbf{B_t}}]^\dagger - Y^\dagger \right) \right\|_F$$

For the first term we have the following perturbation bound [53, 54],

$$\|\hat{\mathbf{O}}_{\mathbf{t}}^{\dagger} - X^{\dagger}\|_F \leq \|\hat{\mathbf{O}}_{\mathbf{t}} - X\|_F \max\{\|X^{\dagger}\|^2, \|\hat{\mathbf{O}}_{\mathbf{t}}^{\dagger}\|^2\} \leq \|\mathcal{N} - \hat{\mathcal{N}}_t\| \sqrt{\frac{10n}{\sigma_n(\mathcal{N})}} \max\{\|X^{\dagger}\|^2, \|\hat{\mathbf{O}}_{\mathbf{t}}^{\dagger}\|^2\}$$

Since we have $\sigma_n(\mathcal{N}) \geq 2\|\mathcal{N} - \hat{\mathcal{N}}\|$, we have $\|\hat{\mathcal{N}}\| \leq 2\|\mathcal{N}\|$ and $2\sigma_n(\hat{\mathcal{N}}) \geq \sigma_n(\mathcal{N})$. Thus,

$$\max\{\|X^{\dagger}\|^2, \|\hat{\mathbf{O}}_{\mathbf{t}}^{\dagger}\|^2\} = \max\left\{\frac{1}{\sigma_n(\mathcal{N})}, \ \frac{1}{\sigma_n(\hat{\mathcal{N}})}\right\} \leq \frac{2}{\sigma_n(\mathcal{N})} \tag{29}$$

Combining these and following the same steps for $\|[\hat{\mathbf{C}}_{\mathbf{F_t}} \ \hat{\mathbf{C}}_{\mathbf{B_t}}]^{\dagger} - Y^{\dagger}\|_F$, we get

$$\left\|\hat{\mathbf{O}}_{\mathbf{t}}^{\dagger} - X^{\dagger}\right\|_F, \ \left\|[\hat{\mathbf{C}}_{\mathbf{F_t}} \ \hat{\mathbf{C}}_{\mathbf{B_t}}]^{\dagger} - Y^{\dagger}\right\|_F \leq \left\|\mathcal{N} - \hat{\mathcal{N}}_t\right\| \sqrt{\frac{40n}{\sigma_n^3(\mathcal{N})}} \tag{30}$$

The following individual bounds obtained by using (29), (30) and triangle inequality:

$$\left\|\left(\hat{\mathbf{O}}_{\mathbf{t}}^{\dagger} - X^{\dagger}\right)\hat{\mathcal{H}}_t^{+}[\hat{\mathbf{C}}_{\mathbf{F_t}} \ \hat{\mathbf{C}}_{\mathbf{B_t}}]^{\dagger}\right\|_F \leq \|\hat{\mathbf{O}}_{\mathbf{t}}^{\dagger} - X^{\dagger}\|_F \|\hat{\mathcal{H}}_t^{+}\| \|[\hat{\mathbf{C}}_{\mathbf{F_t}} \ \hat{\mathbf{C}}_{\mathbf{B_t}}]^{\dagger}\|$$

$$\leq \frac{4\sqrt{5n}\left\|\mathcal{N} - \hat{\mathcal{N}}_t\right\|}{\sigma_n^2(\mathcal{N})}\left(\|\mathcal{H}^{+}\| + \|\hat{\mathcal{H}}_t^{+} - \mathcal{H}^{+}\|\right)$$

$$\left\|X^{\dagger}\left(\hat{\mathcal{H}}_t^{+} - \mathcal{H}^{+}\right)[\hat{\mathbf{C}}_{\mathbf{F_t}} \ \hat{\mathbf{C}}_{\mathbf{B_t}}]^{\dagger}\right\|_F \leq \frac{2\sqrt{n}\|\hat{\mathcal{H}}_t^{+} - \mathcal{H}^{+}\|}{\sigma_n(\mathcal{N})}$$

$$\left\|X^{\dagger}\mathcal{H}^{+}\left([\hat{\mathbf{C}}_{\mathbf{F_t}} \ \hat{\mathbf{C}}_{\mathbf{B_t}}]^{\dagger} - Y^{\dagger}\right)\right\|_F \leq \|X^{\dagger}\| \|\mathcal{H}^{+}\| \|[\hat{\mathbf{C}}_{\mathbf{F_t}} \ \hat{\mathbf{C}}_{\mathbf{B_t}}]^{\dagger} - Y^{\dagger}\|$$

$$\leq \frac{2\sqrt{10n}\left\|\mathcal{N} - \hat{\mathcal{N}}_t\right\|}{\sigma_n^2(\mathcal{N})}\|\mathcal{H}^{+}\|$$

Combining these we get

$$\|\hat{\bar{A}}_t - \mathbf{T}^{\top}\bar{A}\mathbf{T}\|_F \leq \frac{31\sqrt{n}\|\mathcal{H}^{+}\|\left\|\mathcal{N} - \hat{\mathcal{N}}_t\right\|}{2\sigma_n^2(\mathcal{N})} + \|\hat{\mathcal{H}}_t^{+} - \mathcal{H}^{+}\|\left(\frac{4\sqrt{5n}\left\|\mathcal{N} - \hat{\mathcal{N}}_t\right\|}{\sigma_n^2(\mathcal{N})} + \frac{2\sqrt{n}}{\sigma_n(\mathcal{N})}\right)$$

$$\leq \frac{31\sqrt{n}\|\mathcal{H}^{+}\|}{2\sigma_n^2(\mathcal{N})}\left\|\mathcal{N} - \hat{\mathcal{N}}_t\right\| + \frac{13\sqrt{n}}{2\sigma_n(\mathcal{N})}\|\hat{\mathcal{H}}_t^{+} - \mathcal{H}^{+}\|$$

Now consider $\hat{A}_t = \hat{\bar{A}}_t + \hat{F}_t\hat{C}_t$. Using Lemma D.1,

$$\|\hat{A}_t - \mathbf{T}^{\top}\bar{A}\mathbf{T}\|_F$$
$$= \|\hat{\bar{A}}_t + \hat{F}_t\hat{C}_t - \mathbf{T}^{\top}\bar{A}\mathbf{T} - \mathbf{T}^{\top}\bar{F}\bar{C}\mathbf{T}\|_F$$
$$\leq \|\hat{\bar{A}}_t - \mathbf{T}^{\top}\bar{A}\mathbf{T}\|_F + \|(\hat{F}_t - \mathbf{T}^{\top}\bar{F})\hat{C}_t\|_F + \|\mathbf{T}^{\top}\bar{F}(\hat{C}_t - \bar{C}\mathbf{T})\|_F$$
$$\leq \|\hat{\bar{A}}_t - \mathbf{T}^{\top}\bar{A}\mathbf{T}\|_F + \|(\hat{F}_t - \mathbf{T}^{\top}\bar{F})\|_F\|\hat{C}_t - \bar{C}\mathbf{T}\|_F + \|(\hat{F}_t - \mathbf{T}^{\top}\bar{F})\|_F\|\bar{C}\| + \|\bar{F}\|\|(\hat{C}_t - \bar{C}\mathbf{T})\|_F$$
$$\leq \frac{31\sqrt{n}\|\mathcal{H}^{+}\|}{2\sigma_n^2(\mathcal{N})}\left\|\mathcal{N} - \hat{\mathcal{N}}_t\right\| + \frac{13\sqrt{n}}{2\sigma_n(\mathcal{N})}\|\hat{\mathcal{H}}_t^{+} - \mathcal{H}^{+}\| + \frac{10n\|\mathcal{N} - \hat{\mathcal{N}}_t\|^2}{\sigma_n(\mathcal{N})} + (\|\bar{F}\| + \|\bar{C}\|)\|\mathcal{N} - \hat{\mathcal{N}}_t\|\sqrt{\frac{10n}{\sigma_n(\mathcal{N})}}$$
$$\leq \frac{31\sqrt{2nH_e}\|\mathcal{H}\|}{2\sigma_n^2(\mathcal{N})}\|\widehat{\mathcal{G}}_{\mathbf{y}} - \mathcal{G}_{\mathbf{y}}\| + \frac{13\sqrt{nH_e}}{2\sqrt{2}\sigma_n(\mathcal{N})}\|\widehat{\mathcal{G}}_{\mathbf{y}} - \mathcal{G}_{\mathbf{y}}\| + \frac{20nH_e\|\widehat{\mathcal{G}}_{\mathbf{y}} - \mathcal{G}_{\mathbf{y}}\|^2}{\sigma_n(\mathcal{N})}$$
$$\quad + (\|\bar{F}\| + \|\bar{C}\|)\|\widehat{\mathcal{G}}_{\mathbf{y}} - \mathcal{G}_{\mathbf{y}}\|\sqrt{\frac{20nH_e}{\sigma_n(\mathcal{N})}}$$

$$\square$$

## D.4 Bound on the Markov Parameters Estimation Errors

Finally, we will consider the Markov parameter estimates that is constructed by using the parameter estimates. From Theorem 4, for some unitary matrix $\mathbf{T}$, we denote $\Delta A := \|\widehat{A}_t - \mathbf{T}^\top A \mathbf{T}\|$, $\Delta B := \|\widehat{B}_t - \mathbf{T}^\top B\| = \|\widehat{C}_t - C\mathbf{T}\|$. Let $T_A = T_{\mathcal{G}_\mathbf{y}} \dfrac{4c_1^2 \left( \frac{\sqrt{nH_e}(\|\mathcal{H}\| + \sigma_n(\mathcal{H}))}{\sigma_n^2(\mathcal{H})} \right)^2}{(1-\rho(A))^2}$. For $t > \max\{T_A, T_B\}$, $\Delta A \leq \frac{1-\rho(A)}{2}$ and $\Delta B \leq 1$. Using this fact, we have

$$\sum_{j \geq 1}^{H} \|\widehat{C}_t \widehat{A}_t^{j-1} \widehat{B}_t - CA^{j-1}B\|$$

$$\leq \Delta B(\|B\| + \|C\| + 1) + \sum_{i=1}^{H-1} \Phi(A)\rho^i(A)\Delta B(\|B\| + \|C\| + 1) + \|\widehat{A}_t^i - \mathbf{T}^\top A^i \mathbf{T}\|(\|C\|\|B\| + \|B\| + \|C\| + 1)$$

$$\leq \left(1 + \frac{\Phi(A)}{1-\rho(A)}\right)\Delta B(\|B\| + \|C\| + 1) + \Delta A(\|C\|\|B\| + \|B\| + \|C\| + 1) \sum_{i=1}^{H-1}\sum_{j=0}^{i-1}\binom{i}{j}\|A^j\|(\Delta A)^{i-1-j}$$

$$\leq \left(1 + \frac{\Phi(A)}{1-\rho(A)}\right)\Delta B(\|B\| + \|C\| + 1)$$

$$+ \Delta A \Phi(A)(\|C\|\|B\| + \|B\| + \|C\| + 1) \sum_{i=1}^{H-1}\sum_{j=0}^{i-1}\binom{i}{j}\rho^j(A)\left(\frac{1-\rho(A)}{2}\right)^{i-1-j}$$

$$\leq \left(1 + \frac{\Phi(A)}{1-\rho(A)}\right)\Delta B(\|B\| + \|C\| + 1) + \frac{2\Delta A \Phi(A)}{1-\rho(A)}(\|C\|\|B\| + \|B\| + \|C\| + 1)\sum_{i=1}^{H-1}\left[\left(\frac{1+\rho}{2}\right)^i - \rho^i\right]$$

$$\leq \Delta B\left(1 + \frac{\Phi(A)}{1-\rho(A)}\right)(\|B\| + \|C\| + 1) + \frac{2\Delta A \Phi(A)}{(1-\rho(A))^2}(\|C\|\|B\| + \|B\| + \|C\| + 1)$$

$$\gamma_\mathbf{G} = (\|B\| + \|C\| + 1)\left(1 + \frac{\Phi(A)}{1-\rho(A)} + \frac{2\Phi(A)}{(1-\rho(A))^2}\right) + \frac{2\Phi(A)}{(1-\rho(A))^2}\|C\|\|B\|$$

Assuming that $\|F\| + \|C\| > 1$ for simplicity, from the exact expressions of Theorem 4, we have $\Delta A > \Delta B$. For the given $\gamma_\mathbf{G}$ and $\gamma_\mathcal{H}$, we can upper bound the last expression above as follow,

$$\sum_{j \geq 1}^{H} \|\widehat{C}_t \widehat{A}_t^{j-1} \widehat{B}_t - CA^{j-1}B\| \leq \gamma_\mathbf{G}\Delta A \leq \frac{c_1 \gamma_\mathbf{G}\gamma_\mathcal{H}\kappa_e}{\sigma_\star \sqrt{t}}, \tag{31}$$

for

$$\gamma_\mathbf{G} := (\|B\| + \|C\| + 1)\left(1 + \frac{\Phi(A)}{1-\rho(A)} + \frac{2\Phi(A)}{(1-\rho(A))^2}\right) + \frac{2\Phi(A)}{(1-\rho(A))^2}\|C\|\|B\|, \tag{32}$$

$$\kappa_e := \sqrt{m\|C\Sigma C^\top + \sigma_z^2 I\|\left(\log(1/\delta) + \frac{H_e(m+p)}{2}\log\left(\frac{\lambda(m+p) + T\Upsilon^2}{\lambda(m+p)}\right)\right)} + S\sqrt{\lambda} + \frac{\sqrt{H_e}}{T}, \tag{33}$$

$$\gamma_\mathcal{H} := \frac{\sqrt{nH_e}(\|\mathcal{H}\| + \sigma_n(\mathcal{H}))}{\sigma_n^2(\mathcal{H})}. \tag{34}$$

The proof of Theorem 1 is completed by noticing that $\|\widehat{\mathbf{G}}(H) - \mathbf{G}(H)\| = \|[\widehat{G}^{[1]}\ \widehat{G}^{[2]}\ \dots\ \widehat{G}^{[H]}] - [G^{[1]}\ G^{[2]}\ \dots\ G^{[H]}] \leq \sqrt{\sum_{i=1}^{H}\|\widehat{G}^{[i]} - G^{[i]}\|^2}$.

## E Persistence of Excitation

In this section, we provide the persistence of excitation of ADAPTON inputs that is required for consistent estimation of system parameters as pointed out in Theorem 1. We will first consider

open-loop persistence excitation and introduce truncated open-loop noise evolution parameter $\mathcal{G}^{ol}$, Appendix E.1. It represents the effect of noises in the system on the outputs. We will define $\mathcal{G}^{ol}$ for $2H_e$ time steps back in time and show that last $2H_e$ process and measurement noises provide sufficient persistent excitation for the covariates in the estimation problem. Let $\sigma_o < \sigma_{\min}(\mathcal{G}^{ol})$. We will show that there exists a positive $\sigma_o$, *i.e.*, $\mathcal{G}^{ol}$ is full row rank. In the following, $\bar{\phi}_t = P\phi_t$ for a permutation matrix $P$ that gives

$$\bar{\phi}_t = \begin{bmatrix} y_{t-1}^\top & u_{t-1}^\top \ldots y_{t-H}^\top & u_{t-H}^\top \end{bmatrix}^\top \in \mathbb{R}^{(m+p)H}.$$

We assume that, throughout the interaction with the system, the agent has access to a convex compact set of DFCs, $\mathcal{M}$ which is an $r$-expansion of $\mathcal{M}_\psi$, such that $\kappa_\mathcal{M} = \kappa_\psi + r$ and all controllers $\mathbf{M} \in \mathcal{M}$ are persistently exciting the system $\Theta$. In Appendix E.2, we formally define the persistence of excitation condition for the given set $\mathcal{M}$. Finally, in Appendix E.3, we show that persistence of excitation is achieved by the policies that ADAPTON deploys.

### E.1 Persistence of Excitation in Warm-up

Recall the state-space form of the system,

$$
\begin{aligned}
x_{t+1} &= Ax_t + Bu_t + w_t \\
y_t &= Cx_t + z_t.
\end{aligned}
\tag{35}
$$

During the warm-up period, $t \leq T_w$, the input to the system is $u_t \sim \mathcal{N}(0, \sigma_u^2 I)$. Let $f_t = [y_t^\top u_t^\top]^\top$. From the evolution of the system with given input we have the following:

$$f_t = \mathbf{G^o} \begin{bmatrix} w_{t-1}^\top & z_t^\top & u_t^\top & \ldots & w_{t-H}^\top & z_{t-H+1}^\top & u_{t-H+1}^\top \end{bmatrix}^\top + \mathbf{r_t^o}$$

where

$$\mathbf{G^o} := \begin{bmatrix} 0_{m\times n} & I_{m\times m} & 0_{m\times p} & C & 0_{m\times m} & CB & CA & 0_{m\times m} & CAB & \ldots & CA^{H_e-2} & 0_{m\times m} & CA^{H_e-2}B \\ 0_{p\times n} & 0_{p\times m} & I_{p\times p} & 0_{p\times n} & 0_{p\times m} & 0_{p\times p} & 0_{p\times n} & 0_{p\times m} & 0_{p\times p} & \ldots & 0_{p\times n} & 0_{p\times m} & 0_{p\times p} \end{bmatrix}$$

(36)

and $\mathbf{r_t^o}$ is the residual vector that represents the effect of $[w_{i-1}\ z_i\ u_i]$ for $0 \leq i < t - H_e$, which are independent. Notice that $\mathbf{G^o}$ is full row rank even for $H_e = 1$, due to first $(m+p) \times (m+n+p)$ block. Using this, we can represent $\bar{\phi}_t$ as follows

$$
\bar{\phi}_t = \underbrace{\begin{bmatrix} f_{t-1} \\ \vdots \\ f_{t-H_e} \end{bmatrix}}_{\mathbb{R}^{(m+p)H_e}} + \begin{bmatrix} \mathbf{r_{t-1}^o} \\ \vdots \\ \mathbf{r_{t-H_e}^o} \end{bmatrix} = \mathcal{G}^{ol} \underbrace{\begin{bmatrix} w_{t-2} \\ z_{t-1} \\ u_{t-1} \\ \vdots \\ w_{t-2H_e-1} \\ z_{t-2H_e} \\ u_{t-2H_e} \end{bmatrix}}_{\mathbb{R}^{2(n+m+p)H_e}} + \begin{bmatrix} \mathbf{r_{t-1}^o} \\ \vdots \\ \mathbf{r_{t-H_e}^o} \end{bmatrix} \quad \text{where}
$$

$$
\mathcal{G}^{ol} := \begin{bmatrix} \begin{bmatrix} & \mathbf{G^o} & \end{bmatrix} & 0_{(m+p)\times(m+n+p)} & 0_{(m+p)\times(m+n+p)} & 0_{(m+p)\times(m+n+p)} & \cdots \\ 0_{(m+p)\times(m+n+p)} & \begin{bmatrix} & \mathbf{G^o} & \end{bmatrix} & 0_{(m+p)\times(m+n+p)} & 0_{(m+p)\times(m+n+p)} & \cdots \\ & & \ddots & & \\ 0_{(m+p)\times(m+n+p)} & 0_{(m+p)\times(m+n+p)} & \cdots & \begin{bmatrix} & \mathbf{G^o} & \end{bmatrix} & 0_{(m+p)\times(m+n+p)} \\ 0_{(m+p)\times(m+n+p)} & 0_{(m+p)\times(m+n+p)} & 0_{(m+p)\times(m+n+p)} & \cdots & \begin{bmatrix} & \mathbf{G^o} & \end{bmatrix} \end{bmatrix}.
$$

(37)

During warm-up period, from Lemma D.1 of Lale et al. [9], we have that for all $1 \leq t \leq T_w$, with probability $1 - \delta/2$,

$$\|x_t\| \leq X_w := \frac{(\sigma_w + \sigma_u\|B\|)\Phi(A)\rho(A)}{\sqrt{1-\rho(A)^2}}\sqrt{2n\log(12nT_w/\delta)},\tag{38}$$

$$\|z_t\| \leq Z := \sigma_z\sqrt{2m\log(12mT_w/\delta)},\tag{39}$$

$$\|u_t\| \leq U_w := \sigma_u\sqrt{2p\log(12pT_w/\delta)},\tag{40}$$

$$\|y_t\| \leq \|C\|X_w + Z.\tag{41}$$

Thus, during the warm-up phase, we have $\max_{i \leq t \leq T_w} \|\phi_i\| \leq \Upsilon_w \sqrt{H_e}$, where $\Upsilon_w = \|C\|X_w + Z + U_w$. Define

$$T_o = \frac{32\Upsilon_w^4 \log^2\left(\frac{2H_e(m+p)}{\delta}\right)}{\sigma_{\min}^4(\mathcal{G}^{ol})\min\{\sigma_w^4, \sigma_z^4, \sigma_u^4\}}.$$

We have the following lemma that provides the persistence of excitation of inputs in the warm-up period.

**Lemma E.1.** *If the warm-up duration $T_w \geq T_o$, then for $T_o \leq t \leq T_w$, with probability at least $1 - \delta$ we have*

$$\sigma_{\min}\left(\sum_{i=1}^t \phi_i\phi_i^\top\right) \geq t\frac{\sigma_o^2 \min\{\underline{\sigma}_w^2, \underline{\sigma}_z^2, \sigma_u^2\}}{2}. \tag{42}$$

*Proof.* Let $\bar{\mathbf{0}} = 0_{(m+p)\times(m+n+p)}$. Since each block row is full row-rank, we get the following decomposition using QR decomposition for each block row:

$$\mathcal{G}^{ol} = \underbrace{\begin{bmatrix} Q^o & 0_{m+p} & 0_{m+p} & 0_{m+p} & \cdots \\ 0_{m+p} & Q^o & 0_{m+p} & 0_{m+p} & \cdots \\ & & \ddots & & \\ 0_{m+p} & 0_{m+p} & \cdots & Q^o & 0_{m+p} \\ 0_{m+p} & 0_{m+p} & 0_{m+p} & \cdots & Q^o \end{bmatrix}}_{\mathbb{R}^{(m+p)H \times (m+p)H}} \underbrace{\begin{bmatrix} R^o & \bar{\mathbf{0}} & \bar{\mathbf{0}} & \bar{\mathbf{0}} & \cdots \\ \bar{\mathbf{0}} & R^o & \bar{\mathbf{0}} & \bar{\mathbf{0}} & \cdots \\ & & \ddots & & \\ \bar{\mathbf{0}} & \bar{\mathbf{0}} & \cdots & R^o & \bar{\mathbf{0}} \\ \bar{\mathbf{0}} & \bar{\mathbf{0}} & \bar{\mathbf{0}} & \cdots & R^o \end{bmatrix}}_{\mathbb{R}^{(m+p)H \times 2(m+n+p)H}}$$

where $R^o = \begin{bmatrix} \times & \times & \times & \times & \times & \times & \cdots \\ 0 & \times & \times & \times & \times & \times & \cdots \\ \ddots & & & & & & \\ 0 & 0 & 0 & \times & \times & \times & \cdots \end{bmatrix} \in \mathbb{R}^{(m+p)\times H(m+n+p)}$ where the elements in the diagonal are positive numbers. Notice that the first matrix with $Q^0$ is full rank. Also, all the rows of second matrix are in row echelon form and second matrix is full row-rank. Thus, we can deduce that $\mathcal{G}^{ol}$ is full row-rank. Since $\mathcal{G}^{ol}$ is full row rank, we have that

$$\mathbb{E}[\bar{\phi}_t\bar{\phi}_t^\top] \succeq \mathcal{G}^{ol}\Sigma_{w,z,u}\mathcal{G}^{ol\top}$$

where $\Sigma_{w,z,u} \in \mathbb{R}^{2(n+m+p)H \times 2(n+m+p)H} = \mathrm{diag}(\sigma_w^2, \sigma_z^2, \sigma_u^2, \ldots, \sigma_w^2, \sigma_z^2, \sigma_u^2)$. This gives us

$$\sigma_{\min}(\mathbb{E}[\bar{\phi}_t\bar{\phi}_t^\top]) \geq \sigma_{\min}^2(\mathcal{G}^{ol})\min\{\sigma_w^2, \sigma_z^2, \sigma_u^2\}$$

for $t \leq T_w$. As given in (38)-(41), we have that $\|\phi_t\| \leq \Upsilon_w\sqrt{H_e}$ with probability at least $1 - \delta/2$. Given this holds, one can use Theorem 6, to obtain the following which holds with probability $1 - \delta/2$:

$$\lambda_{\max}\left(\sum_{i=1}^t \phi_i\phi_i^\top - \mathbb{E}[\phi_i\phi_i^\top]\right) \leq 2\sqrt{2t}\Upsilon_w^2 H_e\sqrt{\log\left(\frac{2H_e(m+p)}{\delta}\right)}.$$

Using Weyl's inequality, during the warm-up period with probability $1 - \delta$, we have

$$\sigma_{\min}\left(\sum_{i=1}^t \phi_i\phi_i^\top\right) \geq t\sigma_o^2 \min\{\sigma_w^2, \sigma_z^2, \sigma_u^2\} - 2\sqrt{2t}\Upsilon_w^2 H_e\sqrt{\log\left(\frac{2H_e(m+p)}{\delta}\right)}.$$

For all $t \geq T_o := \frac{32\Upsilon_w^4 H_e^2 \log\left(\frac{2H_e(m+p)}{\delta}\right)}{\sigma_o^4 \min\{\sigma_w^4, \sigma_z^4, \sigma_u^4\}}$, we have the stated lower bound. $\square$

Combining Lemma E.1 with Theorem 3 gives

$$\|\widehat{\mathcal{G}}_{\mathbf{y},\mathbf{1}} - \mathcal{G}_{\mathbf{y}}\| \leq \frac{\kappa_e}{\sigma_o\sqrt{T_w}\sqrt{\frac{\min\{\underline{\sigma}_w^2, \underline{\sigma}_z^2, \sigma_u^2\}}{2}}},$$

at the end of warm-up, with probability at least $1 - 2\delta$, in parallel with Appendix D.2.

## E.2 Persistence of Excitation Condition of $\mathrm{M} \in \mathcal{M}$

In this section, we will provide the precise condition of the DFC policies in $\mathcal{M}$, which provides the persistence of excitation if the underlying system is known while designing the control input via any DFC policy in $\mathcal{M}$. The condition is given in (43). Note that in the controller design of ADAPTON, we don't have access to the actual system. In the next section, we show that even though we have errors in the estimates, if the errors are small enough, we can still have persistence of excitation in the inputs.

Now consider when the underlying system is known. If that's the case, the following are the inputs and outputs of the system:

$$u_t = \sum_{j=0}^{H'-1} M_t^{[j]} b_{t-j}(\mathbf{G})$$

$$y_t = [G^{[0]}\ G^{[1]} \ldots\ G^{[H]}] \left[u_t^\top\ u_{t-1}^\top \ldots u_{t-H}^\top\right]^\top + b_t(\mathbf{G}) + \mathbf{r_t}$$

where $\mathbf{r_t} = \sum_{k=H+1}^{t-1} G^{[k]} u_{t-k}$. For $H_e$ defined in Section 4.2, $H_e \geq \max\{2n + 1, \frac{\log(c_H T^2 \sqrt{m}/\sqrt{\lambda})}{\log(1/\upsilon)}\}$, define

$$\phi_t = \left[y_{t-1}^\top \ldots y_{t-H_e}^\top\ u_{t-1}^\top \ldots\ u_{t-H_e}^\top\right]^\top \in \mathbb{R}^{(m+p)H_e}.$$

We have the following decompositions for $\phi_t$:

$$\phi_t = \underbrace{\begin{bmatrix} G^{[0]} & G^{[1]} & \cdots & \cdots & \cdots & G^{[H]} & 0_{m\times p} & 0_{m\times p} & \cdots & 0_{m\times p} \\ 0_{m\times p} & G^{[0]} & \cdots & \cdots & \cdots & G^{[H-1]} & G^{[H]} & 0_{m\times p} & \cdots & 0_{m\times p} \\ & & \ddots & & & & & \ddots & & \\ 0_{m\times p} & \cdots & 0_{m\times p} & G^{[0]} & G^{[1]} & \cdots & \cdots & \cdots & G^{[H-1]} & G^{[H]} \\ I_{p\times p} & 0_{p\times p} & 0_{p\times p} & 0_{p\times p} & 0_{p\times p} & 0_{p\times p} & \cdots & \cdots & \cdots & 0_{p\times p} \\ 0_{p\times p} & I_{p\times p} & 0_{p\times p} & 0_{p\times p} & 0_{p\times p} & 0_{p\times p} & \cdots & \cdots & \cdots & 0_{p\times p} \\ & & \ddots & & & & & & & \\ 0_{p\times p} & 0_{p\times p} & \cdots & I_{p\times p} & 0_{p\times p} & \cdots & \cdots & \cdots & \cdots & 0_{p\times p} \end{bmatrix}}_{\mathcal{T}_\mathbf{G} \in \mathbb{R}^{H_e(m+p)\times(H_e+H)p}} \begin{bmatrix} u_{t-1} \\ \vdots \\ u_{t-H} \\ \vdots \\ u_{t-H-H_e} \\ \mathcal{U}_t \end{bmatrix} + \underbrace{\begin{bmatrix} b_{t-1} \\ \vdots \\ b_{t-H_e} \\ 0_p \\ \vdots \\ 0_p \end{bmatrix}}_{B_y(\mathbf{G})(t)} + \underbrace{\begin{bmatrix} \mathbf{r_{t-1}} \\ \vdots \\ \mathbf{r_{t-H_e}} \\ 0_p \\ \vdots \\ 0_p \end{bmatrix}}_{\mathbf{R}_t}$$

$$\mathcal{U}_t = \underbrace{\begin{bmatrix} M_{t-1}^{[0]} & M_{t-1}^{[1]} & \cdots & \cdots & M_{t-1}^{[H'-1]} & 0_{p\times m} & 0_{p\times m} & \cdots & 0_{p\times m} \\ 0_{p\times m} & M_{t-2}^{[0]} & \cdots & \cdots & M_{t-2}^{[H'-2]} & M_{t-2}^{[H'-1]} & 0_{p\times m} & \cdots & 0_{p\times m} \\ & & \ddots & & & & & \ddots & \\ 0_{p\times m} & \cdots & 0_{p\times m} & M_{t-H_e-H}^{[0]} & \cdots & \cdots & \cdots & \cdots & M_{t-H_e-H}^{[H'-1]} \end{bmatrix}}_{\mathcal{T}_{\mathbf{M}_t} \in \mathbb{R}^{(H_e+H)p\times m(H+H'+H_e-1)}} \underbrace{\begin{bmatrix} b_{t-1}(\mathbf{G}) \\ b_{t-2}(\mathbf{G}) \\ \vdots \\ b_{t-H'+1}(\mathbf{G}) \\ \vdots \\ b_{t-H_e-H-H'+1}(\mathbf{G}) \end{bmatrix}}_{B(\mathbf{G})(t)}$$

$$B(\mathbf{G})(t) = \underbrace{\begin{bmatrix} I_m & 0_m & \cdots & 0_m & C & CA & \cdots & \cdots & \cdots & CA^{t-3} \\ 0_m & I_m & & 0_m & 0_{m\times n} & C & \cdots & \cdots & \cdots & CA^{t-4} \\ & & \ddots & & & \ddots & \ddots & & & \\ 0_m & 0_m & \cdots & I_m & 0_{m\times n} & \cdots & \cdots & C & \cdots & CA^{t-H_e-H-H'-1} \end{bmatrix}}_{\mathcal{O}_t} \underbrace{\begin{bmatrix} z_{t-1} \\ z_{t-2} \\ \vdots \\ z_{t-H_e-H-H'+1} \\ w_{t-2} \\ w_{t-3} \\ \vdots \\ w_1 \end{bmatrix}}_{\eta_t}$$

and $B_y(\mathbf{G})(t) = \underbrace{\begin{bmatrix} I_m & 0_m & \dots & \dots & 0_m & C & \dots & \dots & \dots & CA^{t-3} \\ & \ddots & & & \vdots & & \ddots & & \ddots & \\ 0_m & \dots & I_m & \dots & 0_m & 0_{m \times n} & \dots & C & \dots & CA^{t-H_e-2} \\ & & \mathbf{0}_{(pH_e) \times ((H_e+H+H'-1)m+(t-2)n)} & & & & & & & \end{bmatrix}}_{\bar{\mathcal{O}}_t} \boldsymbol{\eta}_t.$

Combining all gives

$$\phi_t = \left( \mathcal{T}_{\mathbf{G}} \mathcal{T}_{\mathbf{M}_t} \mathcal{O}_t + \bar{\mathcal{O}}_t \right) \boldsymbol{\eta}_t + \mathbf{R}_t.$$

---

**Persistence of Excitation of $\mathbf{M} \in \mathcal{M}\,(H', \kappa_{\mathcal{M}})$ on System $\Theta$.** *For the given system $\Theta$, for $t \geq H + H' + H_e$, $\mathcal{T}_{\mathbf{G}} \mathcal{T}_{\mathbf{M}_t} \mathcal{O}_t + \bar{\mathcal{O}}_t$ is full row rank for all $\mathbf{M} \in \mathcal{M}$, i.e.,*

$$\sigma_{\min}(\mathcal{T}_{\mathbf{G}} \mathcal{T}_{\mathbf{M}_t} \mathcal{O}_t + \bar{\mathcal{O}}_t) > \sigma_c > 0. \tag{43}$$

---

### E.3 Persistence of Excitation in Adaptive Control Period

In this section, we show that the Markov parameter estimates of ADAPTON are well-refined that, the controller of ADAPTON constructed by using a DFC policy in $\mathcal{M}$ still provides persistence of excitation. In other words, we will show that the inaccuracies in the model parameter estimates do not cause lack of persistence of excitation in adaptive control period.

First we have the following lemma, that shows inputs have persistence of excitation during the adaptive control period. Let $d = \min\{m, p\}$. Using (32) and (34), define

$$T_{\epsilon \mathbf{G}} = 4c_1^2 \kappa_{\mathcal{M}}^2 \kappa_{\mathbf{G}}^2 \gamma_{\mathbf{G}}^2 \gamma_{\mathcal{H}}^2 T_{\mathcal{G}_{\mathbf{y}}} \quad T_{cl} = \frac{T_{\epsilon \mathbf{G}}}{\left( \frac{3\sigma_c^2 \min\{\underline{\sigma}_w^2, \underline{\sigma}_z^2\}}{8\kappa_u^2 \kappa_y H_e} - \frac{1}{10T} \right)^2},$$

$$T_c = \frac{2048 \Upsilon_c^4 H_e^2 \log\left( \frac{H_e(m+p)}{\delta} \right) + H'mp \log\left( \kappa_{\mathcal{M}} \sqrt{d} + \frac{2}{\epsilon} \right)}{\sigma_c^4 \min\{\sigma_w^4, \sigma_z^4\}}.$$

for

$$\epsilon = \min\left\{ 1, \frac{\sigma_c^2 \min\{\underline{\sigma}_w^2, \underline{\sigma}_z^2\} \sqrt{\min\{m, p\}}}{68\kappa_b^3 \kappa_{\mathbf{G}} H_e \left( 2\kappa_{\mathcal{M}}^2 + 3\kappa_{\mathcal{M}} + 3 \right)} \right\}$$

**Lemma E.2.** *After $T_c$ time steps in the adaptive control period, with probability $1 - 3\delta$, we have persistence of excitation for the remainder of adaptive control period,*

$$\sigma_{\min} \left( \sum_{i=1}^{t} \phi_i \phi_i^{\top} \right) \geq t \frac{\sigma_c^2 \min\{\sigma_w^2, \sigma_z^2\}}{16}. \tag{44}$$

*Proof.* During the adaptive control period, at time $t$, the input of ADAPTON is given by

$$u_t = \sum_{j=0}^{H'-1} M_t^{[j]} b_{t-j}(\mathbf{G}) + M_t^{[j]} \left( b_{t-j}(\widehat{\mathbf{G}}_i) - b_{t-j}(\mathbf{G}) \right)$$

where

$$b_{t-j}(\mathbf{G}) = y_{t-j} - \sum_{k=1}^{t-j-1} G^{[k]} u_{t-j-k} = z_{t-j} + \sum_{k=1}^{t-j-1} CA^{t-j-k-1} w_k \tag{45}$$

$$b_{t-j}(\widehat{\mathbf{G}}_i) = y_{t-j} - \sum_{k=1}^{H} \widehat{G}_i^{[k]} u_{t-j-k} \tag{46}$$

Thus, we obtain the following for $u_t$ and $y_t$,

$$u_t = \sum_{j=0}^{H'-1} M_t^{[j]} b_{t-j}(\mathbf{G}) + \underbrace{\sum_{j=0}^{H'-1} M_t^{[j]} \left( \sum_{k=1}^{t-j-1} [G^{[k]} - \widehat{G}_i^{[k]}] u_{t-j-k} \right)}_{u_{\Delta b}(t)}$$

$$y_t = [G^{[0]} \ G^{[1]} \ldots \ G^{[H]}] \left[u_t^\top \ u_{t-1}^\top \ldots u_{t-H}^\top\right]^\top + b_t(\mathbf{G}) + \mathbf{r_t}$$

where $\mathbf{r_t} = \sum_{k=H+1}^{t-1} G^{[k]} u_{t-k}$ and $\sum_{k=H}^{t-1} \|G^{[k]}\| \le \psi_{\mathbf{G}}(H+1) \le 1/10T$ which is bounded by the assumption. Notice that $\|u_{\Delta b}(t)\| \le \kappa_{\mathcal{M}} \kappa_u \epsilon_{\mathbf{G}}(1,\delta)$ for all $t \in T_w$. Using the definitions from Appendix E.2, $\phi_t$ can be written as,

$$\phi_t = \left(\mathcal{T}_{\mathbf{G}} \mathcal{T}_{\mathbf{M}_t} \mathcal{O}_t + \bar{\mathcal{O}}_t\right) \boldsymbol{\eta}_t + \mathbf{R}_t + \mathcal{T}_{\mathbf{G}} \mathcal{U}_{\Delta b}(t) \tag{47}$$

where

$$\mathcal{U}_{\Delta b}(t) = \begin{bmatrix} u_{\Delta b}(t-1) \\ u_{\Delta b}(t-2) \\ \vdots \\ u_{\Delta b}(t-H_e) \\ \vdots \\ u_{\Delta b}(t-H_e-H) \end{bmatrix}.$$

Consider the following,

$$\mathbb{E}\left[\phi_t \phi_t^\top\right] = \mathbb{E}\left[ \left(\mathcal{T}_{\mathbf{G}} \mathcal{T}_{\mathbf{M}_t} \mathcal{O}_t + \bar{\mathcal{O}}_t\right) \boldsymbol{\eta}_t \boldsymbol{\eta}_t^\top \left(\mathcal{T}_{\mathbf{G}} \mathcal{T}_{\mathbf{M}_t} \mathcal{O}_t + \bar{\mathcal{O}}_t\right)^\top + \boldsymbol{\eta}_t^\top \left(\mathcal{T}_{\mathbf{G}} \mathcal{T}_{\mathbf{M}_t} \mathcal{O}_t + \bar{\mathcal{O}}_t\right)^\top \left(\mathcal{T}_{\mathbf{G}} \mathcal{U}_{\Delta b}(t) + \mathbf{R}_t\right) \right.$$

$$\left. + \left(\mathcal{T}_{\mathbf{G}} \mathcal{U}_{\Delta b}(t) + \mathbf{R}_t\right)^\top \left(\mathcal{T}_{\mathbf{G}} \mathcal{T}_{\mathbf{M}_t} \mathcal{O}_t + \bar{\mathcal{O}}_t\right) \boldsymbol{\eta}_t + \left(\mathcal{T}_{\mathbf{G}} \mathcal{U}_{\Delta b}(t) + \mathbf{R}_t\right)^\top \left(\mathcal{T}_{\mathbf{G}} \mathcal{U}_{\Delta b}(t) + \mathbf{R}_t\right) \right]$$

$$\sigma_{\min}\left(\mathbb{E}\left[\phi_t \phi_t^\top\right]\right) \ge \sigma_c^2 \min\{\underline{\sigma}_w^2, \underline{\sigma}_z^2\}$$
$$- 2\kappa_b \left(\kappa_{\mathcal{M}} + \kappa_{\mathcal{M}} \kappa_{\mathbf{G}} + 1\right) \sqrt{H_e}((1+\kappa_{\mathbf{G}})\kappa_{\mathcal{M}} \kappa_u \epsilon_{\mathbf{G}}(1,\delta)\sqrt{H_e} + \sqrt{H_e}\kappa_u/10T)$$
$$\ge \sigma_c^2 \min\{\underline{\sigma}_w^2, \underline{\sigma}_z^2\} - 2\kappa_u^2 \kappa_y H_e(2\kappa_{\mathbf{G}} \kappa_{\mathcal{M}} \epsilon_{\mathbf{G}}(1,\delta) + 1/10T)$$

Note that for $T_w \ge T_{cl}$, $\epsilon_{\mathbf{G}}(1,\delta) \le \frac{1}{2\kappa_{\mathcal{M}}\kappa_{\mathbf{G}}} \left( \frac{3\sigma_c^2 \min\{\underline{\sigma}_w^2, \underline{\sigma}_z^2\}}{8\kappa_u^2 \kappa_y H_e} - \frac{1}{10T} \right)$ with probability at least $1-2\delta$. Thus, we get

$$\sigma_{\min}\left(\mathbb{E}\left[\phi_t \phi_t^\top\right]\right) \ge \frac{\sigma_c^2}{4} \min\{\underline{\sigma}_w^2, \underline{\sigma}_z^2\}, \tag{48}$$

for all $t \ge T_w$. Using Lemma F.2, we have that for $\Upsilon_c := (\kappa_y + \kappa_u)$, $\|\phi_t\| \le \Upsilon_c \sqrt{H_e}$ with probability at least $1-2\delta$. Therefore, for a chosen $\mathbf{M} \in \mathcal{M}$, using Theorem 6, we have the following with probability $1-3\delta$:

$$\lambda_{\max}\left(\sum_{i=1}^t \phi_i \phi_i^\top - \mathbb{E}[\phi_i \phi_i^\top]\right) \le 2\sqrt{2t}\Upsilon_c^2 H_e \sqrt{\log\left(\frac{H_e(m+p)}{\delta}\right)}. \tag{49}$$

In order to show that this holds for any chosen $\mathbf{M} \in \mathcal{M}$, we adopt a standard covering argument. We know that from Lemma 5.4 of Simchowitz et al. [6], the Euclidean diameter of $\mathcal{M}$ is at most $2\kappa_{\mathcal{M}}\sqrt{\min\{m,p\}}$, i.e. $\|\mathbf{M}_t\|_F \le \kappa_{\mathcal{M}}\sqrt{\min\{m,p\}}$ for all $\mathbf{M}_t \in \mathcal{M}$. Thus, we can upper bound the covering number as follows,

$$\mathcal{N}(B(\kappa_{\mathcal{M}}\sqrt{\min\{m,p\}}), \|\cdot\|_F, \epsilon) \le \left(\kappa_{\mathcal{M}}\sqrt{\min\{m,p\}} + \frac{2}{\epsilon}\right)^{H'mp}.$$

The following holds for all the centers of $\epsilon$-balls in $\|\mathbf{M}_t\|_F$, for all $t \ge T_w$, with probability $1-3\delta$:

$$\lambda_{\max}\left(\sum_{i=1}^t \phi_i \phi_i^\top - \mathbb{E}[\phi_i \phi_i^\top]\right) \le 2\sqrt{2t}\Upsilon_c^2 H_e \sqrt{\log\left(\frac{H_e(m+p)}{\delta}\right) + H'mp\log\left(\kappa_{\mathcal{M}}\sqrt{\min\{m,p\}} + \frac{2}{\epsilon}\right)}.$$
$$\tag{50}$$

Consider all $\mathbf{M}$ in the $\epsilon$-balls, *i.e.* effect of epsilon perturbation in $\|\mathbf{M}\|_F$ sets, using Weyl's inequality we have with probability at least $1 - 3\delta$,

$$\sigma_{\min}\left(\sum_{i=1}^t \phi_i \phi_i^\top\right) \geq t\left(\frac{\sigma_c^2}{4}\min\{\underline{\sigma}_w^2, \underline{\sigma}_z^2\} - \frac{8\kappa_b^3 \kappa_{\mathbf{G}} H_e \epsilon \left(2\kappa_{\mathcal{M}}^2 + 3\kappa_{\mathcal{M}} + 3\right)}{\sqrt{\min\{m,p\}}}\left(1 + \frac{1}{10T}\right)\right)$$

$$- 2\sqrt{2t}\Upsilon_c^2 H_e \sqrt{\log\left(\frac{H_e(m+p)}{\delta}\right) + H'mp\log\left(\kappa_{\mathcal{M}}\sqrt{\min\{m,p\}} + \frac{2}{\epsilon}\right)}.$$

for $\epsilon \leq 1$. Let $\epsilon = \min\left\{1, \frac{\sigma_c^2 \min\{\underline{\sigma}_w^2, \underline{\sigma}_z^2\}\sqrt{\min\{m,p\}}}{68\kappa_b^3 \kappa_{\mathbf{G}} H_e \left(2\kappa_{\mathcal{M}}^2 + 3\kappa_{\mathcal{M}} + 3\right)}\right\}$. For this choice of $\epsilon$, we get

$$\sigma_{\min}\left(\sum_{i=1}^t \phi_i \phi_i^\top\right) \geq t\left(\frac{\sigma_c^2}{8}\min\{\underline{\sigma}_w^2, \underline{\sigma}_z^2\}\right)$$

$$- 2\sqrt{2t}\Upsilon_c^2 H_e \sqrt{\log\left(\frac{H_e(m+p)}{\delta}\right) + H'mp\log\left(\kappa_{\mathcal{M}}\sqrt{\min\{m,p\}} + \frac{2}{\epsilon}\right)}.$$

For picking $T_w \geq T_c$, we can guarantee that after $T_c$ time steps in the first epoch we have the advertised lower bound. $\qquad\square$

Combining Lemma E.2 with Theorem 3 gives

$$\|\widehat{\mathcal{G}}_{\mathbf{y},\mathbf{i}} - \mathcal{G}_{\mathbf{y}}\| \leq \frac{\kappa_e}{\sigma_c\sqrt{2^{i-1}T_{base}}\sqrt{\frac{\min\{\underline{\sigma}_w^2, \underline{\sigma}_z^2\}}{16}}}, \tag{51}$$

for all $i$, with probability at least $1 - 4\delta$. Setting

$$\sigma_\star^2 := \min\left\{\frac{\sigma_o^2 \underline{\sigma}_w^2}{2}, \frac{\sigma_o^2 \underline{\sigma}_z^2}{2}, \frac{\sigma_o^2 \sigma_y^2}{2}, \frac{\sigma_c^2 \underline{\sigma}_w^2}{16}, \frac{\sigma_c^2 \underline{\sigma}_z^2}{16}\right\},$$

provides the guarantee in Appendix D.2 for warm-up and adaptive control periods.

## F Boundedness Lemmas

**Lemma F.1** (Bounded Nature's $y$). . *For all $t \in [T]$, the following holds with probability at least $1 - \delta$,*

$$\|b_t(\mathbf{G})\| \leq \kappa_b := \overline{\sigma}_z\sqrt{2m\log\frac{6mT}{\delta}} + \|C\|\Phi(A)\sqrt{2n}\left(\rho(A)^t\sqrt{\|\Sigma\|\log\frac{6n}{\delta}} + \frac{\overline{\sigma}_w\sqrt{\log\frac{4nT}{\delta}}}{1-\rho(A)}\right).$$

*Proof.* Using Lemma I.2, the following hold for all $t \in [T]$, with probability at least $1 - \delta$,

$$\|w_t\| \leq \overline{\sigma}_w\sqrt{2n\log\frac{6nT}{\delta}}, \qquad \|z_t\| \leq \overline{\sigma}_z\sqrt{2m\log\frac{6mT}{\delta}}, \qquad \|x_0\| \leq \sqrt{2n\|\Sigma\|\log\frac{6n}{\delta}}. \tag{52}$$

Thus we have,

$$\|b_t(\mathbf{G})\| = \left\|z_t + CA^t x_0 + \sum_{i=0}^{t-1}CA^{t-i-1}w_i\right\| \leq \|z_t\| + \|C\|\left(\|A^t\|\|x_0\| + \left(\max_{1\leq t\leq T}\|w_t\|\right)\sum_{i=0}^\infty \|A^i\|\right)$$

$$\leq \overline{\sigma}_z\sqrt{2m\log\frac{6mT}{\delta}} + \|C\|\Phi(A)\sqrt{2n}\left(\rho(A)^t\sqrt{\|\Sigma\|\log\frac{6n}{\delta}} + \frac{\overline{\sigma}_w\sqrt{\log\frac{4nT}{\delta}}}{1-\rho(A)}\right). \tag{53}$$

$\square$

**Lemma F.2** (Boundedness Lemma). *Let* $\delta \in (0,1)$, $T > T_w \geq T_{\max}$ *and* $\psi_{\mathbf{G}}(H+1) \leq 1/10T$. *For* ADAPTON, *we have the boundedness of the following with probability at least* $1 - 2\delta$:
**Nature's y** : $\|b_t(\mathbf{G})\| \leq \kappa_b, \forall t$,
**Inputs:** $\|u_t\| \leq \kappa_u := 2\max\{\kappa_{u_b}, \kappa_{\mathcal{M}}\kappa_b\}, \forall t$,     **Outputs:** $\|y_t\| \leq \kappa_y := \kappa_b + \kappa_{\mathbf{G}}\kappa_u, \forall t$, *and*
**Nature's y estimates:** $\|b_t(\widehat{\mathbf{G}})\| \leq 2\kappa_b$ , *for all* $t > T_w$.

Proof of this lemma follows similarly from the proof of Lemma 6.1 in Simchowitz et al. [6].

### F.1 Additional Bound on the Markov Parameter Estimates

Define $\alpha$, such that $\alpha \leq \underline{\alpha}_{loss}\left(\sigma_z^2 + \sigma_w^2 \left(\frac{\sigma_{\min}(C)}{1+\|A\|^2}\right)^2\right)$, where right hand side is the effective strong convexity parameter. Define $T_{cx} := T_{\mathcal{G}_{\mathbf{y}}}\frac{16c_1^2\kappa_b^2\kappa_{\mathcal{M}}^2\kappa_{\mathbf{G}}^2 H'\gamma_{\mathbf{G}}^2\gamma_{\mathcal{H}}^2\underline{\alpha}_{loss}}{\alpha}$,    $T_{\epsilon_{\mathbf{G}}} := 4c_1^2\kappa_{\mathcal{M}}^2\kappa_{\mathbf{G}}^2\gamma_{\mathbf{G}}^2\gamma_{\mathcal{H}}^2 T_{\mathcal{G}_{\mathbf{y}}}$ and $T_r = c_1^2\gamma_{\mathbf{G}}^2\gamma_{\mathcal{H}}^2\kappa_{\psi}^2 T_{\mathcal{G}_{\mathbf{y}}}/r^2$.

**Lemma F.3** (Additional Boundedness of Markov Parameter Estimation Error). *Let* $T_w > T_{\max}$, *i.e.* $T_w > \max\{T_{cx}, T_{\epsilon_{\mathbf{G}}}, T_r\}$ *and* $\psi_{\mathbf{G}}(H+1) \leq 1/10T$. *Then*

$$\|\sum_{j\geq 1}\widehat{G}_i^{[j]} - G^{[j]}\| \leq \epsilon_{\mathbf{G}}(i,\delta) \leq \min\left\{\frac{1}{4\kappa_b\kappa_{\mathcal{M}}\kappa_{\mathbf{G}}}\sqrt{\frac{\alpha}{H'\underline{\alpha}_{loss}}}, \frac{1}{2\kappa_{\mathcal{M}}\kappa_{\mathbf{G}}}, \frac{r}{\kappa_{\psi}}\right\}$$

*with probability at least* $1 - 4\delta$, *where* $\epsilon_{\mathbf{G}}(i,\delta) = \frac{2c_1\gamma_{\mathbf{G}}\gamma_{\mathcal{H}}\kappa_e}{\sigma_{\star}\sqrt{2^{i-1}T_{base}}}$.

*Proof.* At the beginning of epoch $i$, using persistence of excitation with high probability in (31), we get

$$\sum_{j\geq 1}^{H}\|\widehat{C}_i\widehat{A}_i^{j-1}\widehat{B}_i - CA^{j-1}B\| \leq \epsilon_{\mathbf{G}}(i,\delta)/2 = \frac{c_1\gamma_{\mathbf{G}}\gamma_{\mathcal{H}}\kappa_e}{\sigma_{\star}\sqrt{2^{i-1}T_{base}}}. \tag{54}$$

From the assumption that $\psi_{\mathbf{G}}(H+1) \leq 1/10T$, we have that $\sum_{j\geq H+1}\|\widehat{G}_1^{[j]} - G^{[j]}\| \leq \epsilon_{\mathbf{G}}(1,\delta)/2$. The second inequality follows from the choice of $T_{\epsilon_{\mathbf{G}}}, T_{cx}$ and $T_r$. $\qquad\square$

## G Proofs for Regret Bound

In order to prove Theorem 2, we follow the proof steps of Theorem 5 of Simchowitz et al. [6]. The main difference is that, ADAPTON updates the Markov parameter estimates in epochs throughout the adaptive control period which provides decrease in the gradient error in each epoch. These updates allow ADAPTON to remove $\mathcal{O}(\sqrt{T})$ term in the regret expression of Theorem 5. In the following, we state how the proof of Theorem 5 of Simchowitz et al. [6] is adapted to the setting of ADAPTON.

**Theorem 5.** *Let* $H'$ *satisfy* $H' \geq 3H \geq 1$, $\psi(\lfloor H'/2 \rfloor - H) \leq \kappa_{\mathcal{M}}/T$ *and* $\psi(H+1) \leq 1/10T$. *If (15) holds for the given setting, after a warm-up period time* $T_w \geq T_{\max}$, *if* ADAPTON *runs with step size* $\eta_t = \frac{12}{\alpha t}$, *then with probability at least* $1 - 5\delta$, *the regret of* ADAPTON *is bounded as*

$$\text{REGRET}(T) \lesssim T_w L\kappa_y^2 + \frac{L^2 H'^3 \min\{m,p\}\kappa_b^4\kappa_{\mathbf{G}}^4\kappa_{\mathcal{M}}^2}{\min\{\alpha, L\kappa_b^2\kappa_{\mathbf{G}}^2\}}\left(1 + \frac{\overline{\alpha}_{loss}}{\min\{m,p\}L\kappa_{\mathcal{M}}}\right)\log\left(\frac{T}{\delta}\right)$$

$$+ \sum_{t=T_w+1}^{T}\epsilon_{\mathbf{G}}^2\left(\lceil\log_2\left(\frac{t}{T_w}\right)\rceil, \delta\right)H'\kappa_b^2\kappa_{\mathcal{M}}^2\left(\frac{\kappa_{\mathbf{G}}^2\kappa_b^2(\overline{\alpha}_{loss}+L)^2}{\alpha} + \kappa_y^2\max\left\{L, \frac{L^2}{\alpha}\right\}\right).$$

*Proof.* Consider the hypothetical "true prediction" y's, $y_t^{pred}$ and losses, $f_t^{pred}(M)$ defined in Definition 8.1 of Simchowitz et al. [6]. Up to truncation by $H$, they describe the true counterfactual output of the system for ADAPTON inputs during the adaptive control period and the corresponding counterfactual loss functions. Lemma I.3, shows that at all epoch $i$, at any time step $t \in [t_i, \ldots, t_{i+1} - 1]$, the gradient $f_t^{pred}(M)$ is close to the gradient of the loss function of ADAPTON:

$$\left\|\nabla f_t\left(\mathbf{M}, \widehat{\mathbf{G}}_i, b_1(\widehat{\mathbf{G}}_i), \ldots, b_t(\widehat{\mathbf{G}}_i)\right) - \nabla f_t^{\text{pred}}(\mathbf{M})\right\|_{\mathrm{F}} \leq C_{\text{approx}}\epsilon_{\mathbf{G}}(i,\delta), \tag{55}$$

where $C_{\text{approx}} := \sqrt{H'}\kappa_{\mathbf{G}}\kappa_{\mathcal{M}}\kappa_b^2\left(16\overline{\alpha}_{loss} + 24L\right)$. For a comparing controller $\mathbf{M}_{comp} \in \mathcal{M}(H', \kappa_{\mathcal{M}})$ and the competing set $\mathcal{M}_{\psi}(H_0', \kappa_{\psi})$, where $\kappa_{\mathcal{M}} = (1+r)\kappa_{\psi}$ and $H_0' = \lfloor \frac{H'}{2} \rfloor - H$, we have the following regret decomposition:

$$
\begin{aligned}
\text{REGRET}(T) \leq &\underbrace{\left(\sum_{t=1}^{T_w} \ell_t\left(y_t, u_t\right)\right)}_{\text{warm-up regret}} + \underbrace{\left(\sum_{t=T_w+1}^{T} \ell_t\left(y_t, u_t\right) - \sum_{t=T_w+1}^{T} F_t^{\text{pred}}\left[\mathbf{M}_{t:t-H}\right]\right)}_{\text{algorithm truncation error}} \\
&+ \underbrace{\left(\sum_{t=T_w+1}^{T} F_t^{\text{pred}}\left[\mathbf{M}_{t:t-H}\right] - \sum_{t=T_w+1}^{T} f_t^{\text{pred}}\left(\mathbf{M}_{comp}\right)\right)}_{f^{\text{pred}} \text{ policy regret}} \\
&+ \underbrace{\left(\sum_{t=T_w+1}^{T} f_t^{\text{pred}}\left(\mathbf{M}_{comp}\right) - \inf_{\mathbf{M}\in\mathcal{M}_{\psi}} \sum_{t=T_w+1}^{T} f_t\left(\mathbf{M}, \mathbf{G}, b_1(\mathbf{G}), \ldots, b_t(\mathbf{G})\right)\right)}_{\text{comparator approximation error}} \\
&+ \underbrace{\left(\inf_{\mathbf{M}\in\mathcal{M}_{\psi}} \sum_{t=T_w+1}^{T} f_t\left(\mathbf{M}, \mathbf{G}, b_1(\mathbf{G}), \ldots, b_t(\mathbf{G})\right) - \inf_{\mathbf{M}\in\mathcal{M}_{\psi}} \sum_{t=T_w+1}^{T} \ell_t\left(y_t^{\mathbf{M}}, u_t^{\mathbf{M}}\right)\right)}_{\text{comparator truncation error}} \\
&+ \underbrace{\left(\inf_{\mathbf{M}\in\mathcal{M}_{\psi}} \sum_{t=1}^{T} \ell_t\left(y_t^{\mathbf{M}}, u_t^{\mathbf{M}}\right) - \sum_{t=0}^{T} \ell(y^{\pi_\star}, u^{\pi_\star})\right)}_{\text{policy approximation error}} \quad (56)
\end{aligned}
$$

Notice that the last term is only required to extend the Theorem 2 to Corollary 6.1. The result of Theorem 2 does not require the last term. We will consider each term separately.

**Warm-up Regret:** From (15) and Lemma F.2, we get $\sum_{t=1}^{T_w} \ell_t\left(y_t, u_t\right) \leq T_w L \kappa_y^2$.

**Algorithm Truncation Error:** From (15), we get

$$
\begin{aligned}
\sum_{t=T_w+1}^{T} \ell_t\left(y_t, u_t\right) - \sum_{t=T_w+1}^{T} F_t^{\text{pred}}\left[\mathbf{M}_{t:t-H}\right] &\leq \sum_{t=T_w+1}^{T} \left|\ell_t\left(y_t, u_t\right) - \ell_t\left(b_t(\mathbf{G}) + \sum_{i=1}^{H} G^{[i]}u_{t-i}, u_t\right)\right| \\
&\leq \sum_{t=T_w+1}^{T} L\kappa_y \left\|y_t - b_t(\mathbf{G}) + \sum_{i=1}^{H} G^{[i]}u_{t-i}\right\| \\
&\leq \sum_{t=T_w+1}^{T} L\kappa_y \left\|\sum_{i=H+1} G^{[i]}u_{t-i}\right\| \\
&\leq TL\kappa_y\kappa_u\psi_{\mathbf{G}}(H+1)
\end{aligned}
$$

Since $\psi_{\mathbf{G}}(H+1) \leq 1/10T$, we get $\sum_{t=T_w+1}^{T} \ell_t\left(y_t, u_t\right) - \sum_{t=T_w+1}^{T} F_t^{\text{pred}}\left[\mathbf{M}_{t:t-H}\right] \leq L\kappa_y\kappa_u/10$.

**Comparator Truncation Error:** Similar to algorithm truncation error above,

$$
\inf_{\mathbf{M}\in\mathcal{M}_{\psi}} \sum_{t=T_w+1}^{T} f_t\left(\mathbf{M}, \mathbf{G}, b_1(\mathbf{G}), \ldots, b_t(\mathbf{G})\right) - \inf_{\mathbf{M}\in\mathcal{M}_{\psi}} \sum_{t=T_w+1}^{T} \ell_t\left(y_t^{\mathbf{M}}, u_t^{\mathbf{M}}\right) \leq TL\kappa_{\mathbf{G}}\kappa_{\mathcal{M}}^2\kappa_b^2\psi_{\mathbf{G}}(H+1)
$$

$$
\leq L\kappa_{\mathbf{G}}\kappa_{\mathcal{M}}^2\kappa_b^2/10
$$

**Policy Approximation Error:** By the assumption that $M_\star$ lives in the given convex set $\mathcal{M}_\psi$ and (15), using Lemma A.1, we get

$$\inf_{\mathbf{M}\in\mathcal{M}_\psi} \sum_{t=1}^T \ell_t\left(y_t^{\mathbf{M}}, u_t^{\mathbf{M}}\right) - \sum_{t=1}^T \ell_t(y_t^{\pi_\star}, u_t^{\pi_\star}) \leq \sum_{t=1}^T \ell_t\left(y_t^{\mathbf{M}_\star}, u_t^{\mathbf{M}_\star}\right) - \ell_t(y_t^{\pi_\star}, u_t^{\pi_\star})$$

$$\leq TL\kappa_y \left(\psi(H_0')\kappa_b + \psi(H_0')\kappa_{\mathbf{G}}\kappa_b\right)$$

$$\leq 2TL\kappa_y\kappa_{\mathbf{G}}\kappa_b\psi(H_0')$$

Since $\psi(H_0') \leq \kappa_{\mathcal{M}}/T$, we get $\inf_{\mathbf{M}\in\mathcal{M}_0} \sum_{t=1}^T \ell_t\left(y_t^{\mathbf{M}}, u_t^{\mathbf{M}}\right) - \sum_{t=1}^T \ell_t(y_t^{\pi_\star}, u_t^{\pi_\star}) \leq 2L\kappa_{\mathcal{M}}\kappa_y\kappa_{\mathbf{G}}\kappa_b$.

$\mathbf{f}^{\mathrm{pred}}$ **Policy Regret** : In order to utilize Theorem 8, we need the strong convexity, Lipschitzness and smoothness properties stated in the theorem. Due to Lemma F.3, Lemmas I.4-I.6 provide those conditions. Combining these with (55), we obtain the following adaptation of Theorem 8:

**Lemma G.1.** *For step size* $\eta = \frac{12}{\alpha t}$, *the following bound holds with probability* $1-\delta$:

$$\mathbf{f}^{\mathrm{pred}} \text{ \textit{policy regret} } + \frac{\alpha}{48} \sum_{t=T_w+1}^T \|\mathbf{M}_t - \mathbf{M}_{comp}\|_F^2$$

$$\lesssim \frac{L^2 H'^3 \min\{m,p\}\kappa_b^4\kappa_{\mathbf{G}}^4\kappa_{\mathcal{M}}^2}{\min\{\alpha, L\kappa_b^2\kappa_{\mathbf{G}}^2\}} \left(1 + \frac{\overline{\alpha}_{loss}}{\min\{m,p\}L\kappa_{\mathcal{M}}}\right) \log\left(\frac{T}{\delta}\right) + \frac{1}{\alpha} \sum_{t=T_w+1}^T C_{approx}^2 \epsilon_{\mathbf{G}}^2 \left(\left\lceil \log_2\left(\frac{t}{T_w}\right)\right\rceil, \delta\right)$$

*Proof.* Let $d = \min\{m,p\}$. We can upper bound the right hand side of Theorem 8 via following proof steps of Theorem 4 of Simchowitz et al. [6]:

$$\mathbf{f}^{\mathrm{pred}}\mathbf{p.r.} - \left(\frac{6}{\alpha}\sum_{t=k+1}^T \|\epsilon_t\|_2^2 - \frac{\alpha}{48}\sum_{t=1}^T \|\mathbf{M}_t - \mathbf{M}_{comp}\|_F^2\right) \lesssim \frac{L^2 H'^3 d\kappa_b^4\kappa_{\mathbf{G}}^4\kappa_{\mathcal{M}}^2}{\min\{\alpha, L\kappa_b^2\kappa_{\mathbf{G}}^2\}}\left(1 + \frac{\overline{\alpha}_{loss}}{dL\kappa_{\mathcal{M}}}\right)\log\left(\frac{T}{\delta}\right)$$

$$\mathbf{f}^{\mathrm{pred}}\mathbf{p.r.} + \frac{\alpha}{48}\sum_{t=1}^T \|\mathbf{M}_t - \mathbf{M}_{comp}\|_F^2 \lesssim \frac{L^2 H'^3 d\kappa_b^4\kappa_{\mathbf{G}}^4\kappa_{\mathcal{M}}^2}{\min\{\alpha, L\kappa_b^2\kappa_{\mathbf{G}}^2\}}\left(1 + \frac{\overline{\alpha}_{loss}}{dL\kappa_{\mathcal{M}}}\right)\log\left(\frac{T}{\delta}\right)$$

$$+ \frac{1}{\alpha}\sum_{t=T_w+1}^T C_{\mathrm{approx}}^2 \epsilon_{\mathbf{G}}^2\left(\left\lceil\log_2\left(\frac{t}{T_w}\right)\right\rceil, \delta\right), \quad (57)$$

where (57) follows from (55). $\qquad\square$

**Comparator Approximation Error:**

**Lemma G.2.** *Suppose that $H' \geq 2H'_0 - 1 + H$, $\psi_{\mathbf{G}}(H+1) \leq 1/10T$. Then for all $\tau > 0$,*

*Comp. app. err.* $\leq 4L\kappa_y\kappa_u\kappa_{\mathcal{M}}$

$$+ \sum_{t=T_w+1}^{T} \left[ \tau \left\| \mathbf{M}_t - \mathbf{M}_{\text{comp}} \right\|_F^2 + 8\kappa_y^2\kappa_b^2\kappa_{\mathcal{M}}^2 (H + H') \max\left\{ L, \frac{L^2}{\tau} \right\} \epsilon_{\mathbf{G}}^2 \left( \left\lceil \log_2\left(\frac{t}{T_w}\right) \right\rceil, \delta \right) \right]$$

*Proof.* The lemma can be proven using the proof of Proposition 8.2 of Simchowitz et al. [6]. Using Lemma E.3 and adapting Lemma E.4 in Simchowitz et al. [6] such that $\mathbf{M}_{comp}^{[i]} = M_*^{[i]}\mathbb{I}_{i \leq H'_0 - 1} + \sum_{a=0}^{H'_0-1}\sum_{b=0}^{H}\sum_{c=0}^{H'_0-1} M_*^{[a]}(\widehat{G}_1^{[b]} - G^{[b]})M_*^{[c]}\mathbb{I}_{a+b+c=i}$ for $\mathbf{M}_* = \arg\min_{\mathbf{M}\in\mathcal{M}_\psi}\sum_{t=T_w+1}^{T} \ell_t(y_t^{\mathbf{M}}, u_t^{\mathbf{M}})$ and due to Lemma F.3 we have $\mathbf{M}_{comp} \in \mathcal{M}$:

$$\sum_{t=T_w+1}^{T} f_t^{\text{pred}}(\mathbf{M}_{comp}) - \inf_{\mathbf{M}\in\mathcal{M}_0} \sum_{t=T_w+1}^{T} f_t(\mathbf{M}, \mathbf{G}, b_1(\mathbf{G}), \ldots, b_t(\mathbf{G}))$$

$$\leq 4L\kappa_y\sum_{t=T_w+1}^{T}\epsilon_{\mathbf{G}}^2\left(\left\lceil\log_2\left(\frac{t}{T_w}\right)\right\rceil, \delta\right)\kappa_{\mathcal{M}}^2\kappa_b\left(\kappa_{\mathcal{M}} + \frac{\kappa_b}{4\tau}\right) + \kappa_u\kappa_{\mathcal{M}}\psi_{\mathbf{G}}(H+1) + (H+H')\tau\left\|\mathbf{M}_t - \mathbf{M}_{\text{comp}}\right\|_F^2$$

$$\leq \sum_{t=T_w+1}^{T}\left[\tau\left\|\mathbf{M}_t - \mathbf{M}_{\text{comp}}\right\|_F^2 + 8\kappa_y^2\kappa_b^2\kappa_{\mathcal{M}}^2(H+H')\max\left\{L, \frac{L^2}{\tau}\right\}\epsilon_{\mathbf{G}}^2\left(\left\lceil\log_2\left(\frac{t}{T_w}\right)\right\rceil, \delta\right)\right]$$

$$+ 4TL\kappa_y\kappa_u\kappa_{\mathcal{M}}\psi_{\mathbf{G}}(H+1)$$

$$\leq 4L\kappa_y\kappa_u\kappa_{\mathcal{M}} + \sum_{t=T_w+1}^{T}\left[\tau\left\|\mathbf{M}_t - \mathbf{M}_{\text{comp}}\right\|_F^2 + 8\kappa_y^2\kappa_b^2\kappa_{\mathcal{M}}^2(H+H')\max\left\{L, \frac{L^2}{\tau}\right\}\epsilon_{\mathbf{G}}^2\left(\left\lceil\log_2\left(\frac{t}{T_w}\right)\right\rceil, \delta\right)\right]$$

$\square$

Combining all the terms bounded above, with $\tau = \frac{\alpha}{48}$ gives

$\text{REGRET}(T)$

$$\lesssim T_wL\kappa_y^2 + L\kappa_y\kappa_u/10 + L\kappa_{\mathbf{G}}\kappa_{\mathcal{M}}^2\kappa_b^2/10 + 2L\kappa_{\mathcal{M}}\kappa_y\kappa_{\mathbf{G}}\kappa_b + 4L\kappa_y\kappa_u\kappa_{\mathcal{M}}$$

$$+ \frac{L^2H'^3\min\{m,p\}\kappa_b^4\kappa_{\mathbf{G}}^4\kappa_{\mathcal{M}}^2}{\min\{\alpha, L\kappa_b^2\kappa_{\mathbf{G}}^2\}}\left(1 + \frac{\overline{\alpha}_{loss}}{\min\{m,p\}L\kappa_{\mathcal{M}}}\right)\log\left(\frac{T}{\delta}\right) + \frac{1}{\alpha}\sum_{t=T_w+1}^{T}C_{\text{approx}}^2\epsilon_{\mathbf{G}}^2\left(\left\lceil\log_2\left(\frac{t}{T_w}\right)\right\rceil, \delta\right)$$

$$+ \sum_{t=T_w+1}^{T} 8\kappa_y^2\kappa_b^2\kappa_{\mathcal{M}}^2(H+H')\max\left\{L, \frac{48L^2}{\alpha}\right\}\epsilon_{\mathbf{G}}^2\left(\left\lceil\log_2\left(\frac{t}{T_w}\right)\right\rceil, \delta\right)$$

$$\lesssim T_wL\kappa_y^2$$

$$+ \frac{L^2H'^3\min\{m,p\}\kappa_b^4\kappa_{\mathbf{G}}^4\kappa_{\mathcal{M}}^2}{\min\{\alpha, L\kappa_b^2\kappa_{\mathbf{G}}^2\}}\left(1 + \frac{\overline{\alpha}_{loss}}{\min\{m,p\}L\kappa_{\mathcal{M}}}\right)\log\left(\frac{T}{\delta}\right)$$

$$+ \sum_{t=T_w+1}^{T}\epsilon_{\mathbf{G}}^2\left(\left\lceil\log_2\left(\frac{t}{T_w}\right)\right\rceil, \delta\right)\left\{\frac{H'\kappa_{\mathbf{G}}^2\kappa_{\mathcal{M}}^2\kappa_b^4(\overline{\alpha}_{loss}+L)^2}{\alpha} + \kappa_y^2\kappa_b^2\kappa_{\mathcal{M}}^2(H+H')\max\left\{L, \frac{48L^2}{\alpha}\right\}\right\}$$

$\square$

Following the doubling update rule of ADAPTON for the epoch lengths, after $T$ time steps of agent-environment interaction, the number of epochs is $\mathcal{O}(\log T)$. From Lemma F.3, at any time step $t$ during the $i$'th epoch, *i.e.*, $t \in [t_i, \ldots, t_i - 1]$, $\epsilon_{\mathbf{G}}^2(i, \delta) = \mathcal{O}(polylog(T)/2^{i-1}T_{base})$. Therefore, update rule of ADAPTON yields,

$$\sum_{t=T_{base}+1}^{T} \epsilon_{\mathbf{G}}^2\left(\left\lceil\log_2\left(\frac{t}{T_w}\right)\right\rceil, \delta\right) = \sum_{i=1}^{\mathcal{O}(\log T)} 2^{i-1}T_{base}\epsilon_{\mathbf{G}}^2(i, \delta) \leq \mathcal{O}(polylog(T)) \quad (58)$$

Using the result of (58), we can bound the third term of the regret upper bound in Theorem 5 with a $polylog(T)$ bound which gives the advertised result in Theorem 2 and using the policy approximation error term we obtain Corollary 6.1.

$\square$

## H   Additional Results

Consider the case where the condition on persistence of excitation of $\mathcal{M}$ does not hold. In order to efficiently learn the model parameters and minimize the regret, one can add an additional independent Gaussian excitation to the control input $u_t$ for each time step $t$. This guarantees the concentration of Markov parameter estimates, but it also results in an extra regret term in the bound of Theorem 5. If the variance of the added Gaussian vector is set to be $\widetilde{\sigma}^2$, exploiting the Lipschitzness of the loss functions, the additive regret of the random excitation is $\tilde{\mathcal{O}}(T\widetilde{\sigma})$. Following the results in Lemma F.3, the additional random excitation helps in parameter estimation and concentration of Markov parameters up to the error of $\mathcal{O}(polylog(T)/\sqrt{\widetilde{\sigma}^2 t})$. Since the contribution of the error in the Markov parameter estimates in the Theorem 5 is quadratic, the contribution of this error in the regret through $R_3$ will be $\mathcal{O}(polylog(T)/\widetilde{\sigma}^2)$.

**Corollary H.1.** *When the condition on persistent excitation of all* $\mathbf{M}$ *is not fulfilled, adding independent Gaussian vectors with variance of* $\mathcal{O}(1/T^{1/3})$ *to the inputs in adaptive control period results in the regret upper bound of* $\tilde{\mathcal{O}}(T^{2/3})$.

## I   Technical Lemmas and Theorems

**Theorem 6** (Matrix Azuma [55])**.** *Consider a finite adapted sequence* $\{\boldsymbol{X}_k\}$ *of self-adjoint matrices in dimension* $d$, *and a fixed sequence* $\{\boldsymbol{A}_k\}$ *of self-adjoint matrices that satisfy*

$$\mathbb{E}_{k-1}\boldsymbol{X}_k = \boldsymbol{0} \text{ and } \boldsymbol{A}_k^2 \succeq \boldsymbol{X}_k^2 \text{ almost surely.}$$

*Compute the variance parameter*

$$\sigma^2 := \left\| \sum_k \boldsymbol{A}_k^2 \right\|$$

*Then, for all* $t \geq 0$

$$\mathbb{P}\left\{ \lambda_{\max}\left( \sum_k \boldsymbol{X}_k \right) \geq t \right\} \leq d \cdot \mathrm{e}^{-t^2/8\sigma^2}$$

**Theorem 7** (Self-normalized bound for vector-valued martingales [56])**.** *Let* $(\mathcal{F}_t; k \geq 0)$ *be a filtration,* $(m_k; k \geq 0)$ *be an* $\mathbb{R}^d$*-valued stochastic process adapted to* $(\mathcal{F}_k)$, $(\eta_k; k \geq 1)$ *be a real-valued martingale difference process adapted to* $(\mathcal{F}_k)$. *Assume that* $\eta_k$ *is conditionally sub-Gaussian with constant R. Consider the martingale*

$$S_t = \sum_{k=1}^{t} \eta_k m_{k-1}$$

*and the matrix-valued processes*

$$V_t = \sum_{k=1}^{t} m_{k-1} m_{k-1}^{\top}, \quad \overline{V}_t = V + V_t, \quad t \geq 0$$

*Then for any* $0 < \delta < 1$, *with probability* $1 - \delta$

$$\forall t \geq 0, \quad \|S_t\|_{V_t^{-1}}^2 \leq 2R^2 \log\left( \frac{\det\left(\overline{V}_t\right)^{1/2} \det(V)^{-1/2}}{\delta} \right)$$

**Theorem 8** (Theorem 8 of Simchowitz et al. [6]). *Suppose that $\mathcal{K} \subset \mathbb{R}^d$ and $h \geq 1$. Let $F_t := \mathcal{K}^{h+1} \to \mathbb{R}$ be a sequence of $L_c$ coordinatewise-Lipschitz functions with the induced unary functions $f_t(x) := F_t(x, \ldots, x)$ which are $L_f$-Lipschitz and $\beta$-smooth. Let $f_{t;k}(x) := \mathbb{E}[f_t(x)|\mathcal{F}_{t-k}]$ be $\alpha$-strongly convex on $\mathcal{K}$ for a filtration $(\mathcal{F}_t)_{t \geq 1}$. Suppose that $z_{t+1} = \Pi_{\mathcal{K}}(z_t - \eta \boldsymbol{g}_t)$, where $\boldsymbol{g}_t = \nabla f_t(z_t) + \epsilon_t$ for $\|\boldsymbol{g}_t\|_2 \leq L_{\mathbf{g}}$, and $\mathrm{Diam}(\mathcal{K}) \leq D$. Let the gradient descent iterates be applied for $t \geq t_0$ for some $t_0 \leq k$, with $z_0 = z_1 = \cdots = z_{t_0} \in \mathcal{K}$ for $k \geq 1$. Then with step size $\eta_t = \frac{3}{\alpha t}$, the following bound holds with probability $1 - \delta$ for all comparators $z_\star \in \mathcal{K}$ simultaneously:*

$$\sum_{t=k+1}^{T} f_t(z_t) - f_t(z_\star) - \left( \frac{6}{\alpha} \sum_{t=k+1}^{T} \|\epsilon_t\|_2^2 - \frac{\alpha}{12} \sum_{t=1}^{T} \|z_t - z_\star\|_2^2 \right)$$

$$\lesssim \alpha k D^2 + \frac{(kL_{\mathrm{f}} + h^2 L_{\mathrm{c}}) L_{\mathrm{g}} + kdL_{\mathrm{f}}^2 + k\beta L_{\mathrm{g}}}{\alpha} \log(T) + \frac{kL_{\mathrm{f}}^2}{\alpha} \log \left( \frac{1 + \log(e + \alpha D^2)}{\delta} \right)$$

**Lemma I.1** (Regularized Design Matrix Lemma [56]). *When the covariates satisfy $\|z_t\| \leq c_m$, with some $c_m > 0$ w.p.1 then*

$$\log \frac{\det(V_t)}{\det(\lambda I)} \leq d \log \left( \frac{\lambda d + tc_m^2}{\lambda d} \right)$$

*where $V_t = \lambda I + \sum_{i=1}^{t} z_i z_i^\top$ for $z_i \in \mathbb{R}^d$.*

**Lemma I.2** (Norm of a subgaussian vector [38]). *Let $v \in \mathbb{R}^d$ be a entry-wise $R$-subgaussian random variable. Then with probability $1 - \delta$, $\|v\| \leq R\sqrt{2d \log(2d/\delta)}$.*

**Lemma I.3** (Lemma 8.1 of Simchowitz et al. [6]). *For any $\mathbf{M} \in \mathcal{M}$, let $f_t^{pred}(\mathbf{M})$ denote the unary counterfactual loss function induced by true truncated counterfactuals (Definition 8.1 of Simchowitz et al. [6]). During the $i$'th epoch of adaptive control period, at any time step $t \in [t_i, \ldots, t_{i+1} - 1]$, for all $i$, we have that*

$$\left\| \nabla f_t \left( \mathbf{M}, \widehat{\mathbf{G}}_i, b_1(\widehat{\mathbf{G}}_i), \ldots, b_t(\widehat{\mathbf{G}}_i) \right) - \nabla f_t^{pred}(\mathbf{M}) \right\|_{\mathrm{F}} \leq C_{approx} \, \epsilon_{\mathbf{G}}(i, \delta),$$

*where $C_{approx} := \sqrt{H'} \kappa_{\mathbf{G}} \kappa_{\mathcal{M}} \kappa_b^2 (16\overline{\alpha}_{loss} + 24L)$.*

**Lemma I.4** (Lemma 8.2 of Simchowitz et al. [6]). *For any $\mathbf{M} \in \mathcal{M}$, $f_t^{pred}(\mathbf{M})$ is $\beta$-smooth, where $\beta = 16H' \kappa_b^2 \kappa_{\mathbf{G}}^2 \overline{\alpha}_{loss}$.*

**Lemma I.5** (Lemma 8.3 of Simchowitz et al. [6]). *For any $\mathbf{M} \in \mathcal{M}$, given $\epsilon_{\mathbf{G}}(i, \delta) \leq \frac{1}{4\kappa_b \kappa_{\mathcal{M}} \kappa_{\mathbf{G}}} \sqrt{\frac{\alpha}{H' \overline{\alpha}_{loss}}}$, conditional unary counterfactual loss function induced by true counterfactuals are $\alpha/4$ strongly convex.*

**Lemma I.6** (Lemma 8.4 of Simchowitz et al. [6]). *Let $L_f = 4L\sqrt{H'} \kappa_b^2 \kappa_{\mathbf{G}}^2 \kappa_{\mathcal{M}}$. For any $\mathbf{M} \in \mathcal{M}$ and for $T_w \geq T_{\max}$, $f_t^{pred}(\mathbf{M})$ is $4L_f$-Lipschitz, $f_t^{pred}[\mathbf{M}_{t:t-H}]$ is $4L_f$ coordinate Lipschitz. Moreover, $\max_{\mathbf{M} \in \mathcal{M}} \left\| \nabla f_t \left( \mathbf{M}, \widehat{\mathbf{G}}_i, b_1(\widehat{\mathbf{G}}_i), \ldots, b_t(\widehat{\mathbf{G}}_i) \right) \right\|_2 \leq 4L_f$.*