[Reviews · NeurIPS 2020]

Review 1

Summary and Contributions: Update: Thanks for the thoughtful response. To clarify, while PE is a common assumption in classical control literature, it is not common in more recent nonasymptotic work.. If one were to assume PE in the state feedback setting, then injecting noise would not be necessary and better regret could be achieved -- but lower bounds tell us that this is not the case. So justifying the applicability of the assumption in this output feedback setting is crucial, and I'm happy to hear that it ends up being a mild assumption satisfied by well known controllers. --- This paper presents a method, AdaptOn for adaptive output-feedback control for linear systems with time-varying convex loss functions. AdaptOn estimates the dynamics each epoch and then runs online gradient descent. The authors show that this method achieves a regret polylogarithmic in T for controllers satisfying a persistence of excitation condition.

Strengths: This work seems to be theoretically sound. The combination of online gradient descent with model estimation is novel, as is the analysis. The main result is significant, by showing that for partially observed systems, a polylog(T) regret can be achieved. This is somewhat surprising, since upper and lower bounds for sqrt(T) regret have been shown in the case that states are fully observed. The analysis includes an interesting argument about persistence of excitation in output feedback problems. This perspective highlights that the existence of measurement noise induces a degree of exploration.

Weaknesses: My main reservation with this paper is that it relies on restricting the controller synthesis to the class of persistently exciting controllers. This class is crucial to the proposed algorithm, since in Corollary 8.1 we see that only a T^2/3 regret bound would follow by these techniques otherwise. However, the condition is not sufficiently explained or motivated in the body of the paper. Is this a natural set of control policies to consider? If it is not, then the regret baseline is less meaningful, and the results are less impressive. It is unclear whether partially observed linear systems admit polylogarithmic regret due to an inherent difference with state observation, or due to artificial restrictions emposed by the persistence of excitation condition on control policies.

Correctness: As far as I can tell, the results seem to be correct.

Clarity: The paper is generally clear and has a well-organized appendix.

Relation to Prior Work: Overall, the paper is well situated with respect to related work. However, a longer and more explicit discussion of the relation to the upper and lower bounds presented in [46] is warranted.

Reproducibility: Yes

Additional Feedback:


Review 2

Summary and Contributions: This work establishes a logarithmic regret bound for the task of controlling an unknown latent-state linear dynamical system subject to stongly convex costs under some conditions. To obtain logarithmic regret, the typical scheme is ensure that the information obtained during exploitation is enough to drive down parameter estimate error. Mania et al established that "E" error in parameters translates to "E^2" increase instantaneous cost (or Simchowitz et al for strongly convex costs). Therefore, establishing "E" scales as T^-0.5 suffices. The latter requirement is met for least squares as long as the covariance of covariates are lower bounded. This latter property, true under appropriate assumptions, is the central observation here. It should be noted that the assumption of persistent excitation here rules out the applicability of the result to LQRs, where T^0.5 is optimal (Simchowitz, Foster).

Strengths: + This work offers the first log regret gurantee for learning partially observable systems. + The closed-loop sys-id procedure, while an adaptation of the known sys-id procedures, completes an improtant part of the picture.

Weaknesses: + While persistent excitation is necessary for logarithmic regret, ideally the algorithm should default to a T^0.5 regret bound in the absence of the former, while offering log regret otherwise. Can the authors say if there are any modifications of the present approach that permit this? + The previous works in this space often did not make an assumption on the stability of the system, but instead operated with stabilizable systems. Note that superimposed stabilizing controller on a stabilizable system is not equivalent to a stable system, because the cost thus must take into account the suggested actions along with the actions of the stabilizing controller. Sometimes considerable effort in the past works has put in this aspect. The absence here limits the applicability. (It is also not apparent if the results extend.)

Correctness: While details were not checked, the claims made look correct.

Clarity: The paper is generally well written.

Relation to Prior Work: To the reviewer's knowledge, the best known bounds (granted with fewer assumption) for this setup scale as T^0.5.

Reproducibility: No

Additional Feedback: Thanks for the response -- the reviewer opts to retain their score.


Review 3

Summary and Contributions: This paper studies the online learning problem for the linear quadratic Gaussian (LQG) model. The proposed algorithm is a combination of System Identification (sys-id) and online convex optimization (OCO) with memory. The proposed algorithm is shown to achieve a polylog(T) regret.

Strengths: This seems the first work that achieves polylog(T) regret for LQG model. LQG is a classical model in the control literature.

Weaknesses: 1. Although the authors claim the algorithm is an RL algorithm, it seems unclear to me how this algorithm can be applied to general reinforcement learning problem. It seems that the algorithm highly rely on the linear dynamical system. If the algorithm and theory is limited only to LQG, it would be less relevant to a machine learning conference. 2. It seems that the proof technic is much dependent on [Simchowitz et al 2020]. The improvement is that the authors propose a new system identification approach and a doubling trick.

Correctness: It seems that the theory is correct. It would be nice to have numerical experiments to validate the polylog T regret.

Clarity: This paper is overall well written. It would be better if the following two points can be made clearer: 1. How does studying LQG help for learning POMDP and can the proposed algorithm be applied to a POMDP? 2. How do the theory and proof differ from [Simchowitz et al 2020]?

Relation to Prior Work: It discussed prior works. However, it would be nice if the most relevant paper, [Simchowitz et al 2020], could be compared in more depth.

Reproducibility: Yes

Additional Feedback: Update after reading the author response: I agree that LQG is a fundamental problem in control theory and establishing a polylogT regret for this problem is significant. But I don't think the proposed algorithm can be modified for POMDP. Thus, it seems a bit oversold by saying that solving LQG is a first step for solving pomdp. As pointed our by another reviewer, it would be nice to highlight that the system is assumed stable. Also I would recommend the authors to highlight technical novelty given [Simchowitz et al 2020] in the revision. Numerical experiments are also helpful to validate the polylog T regret.

[Author Response · NeurIPS 2020]

We thank the reviewers for their effort and insightful comments during these unprecedented times. In this work, we
propose the *first finite-time system identification algorithm for partially observable linear dynamical systems (LDS)*
*in adaptive and closed-loop settings*. Prior estimation methods only work when the actions/controls are iid random
noise and do not allow for any exploitation or strategic exploration. This strong limitation significantly restricted
the regret minimization and algorithm design in LDS with partial observations [3-6,9,12]. Our proposed estimation
algorithm allows the data collection with an adaptive controller and the design of fully adaptive RL methods. We
believe this contribution alone has a great interest in both RL and control communities. Ultimately, we deploy this
estimation method, propose the first "truly" adaptive control algorithm in partially observable LDS, and obtain the *first*
*polylogarithmic regret* in this challenging setting. Our results provide a clear improvement over the prior works and
shed light to further developments in the field.

**Relevance of linear systems to RL and machine learning community:** (**R3**). We would like to highlight that the
setting of our work is more general than classical LQG since our algorithm can handle time-varying and adversarial
cost functions that subsume LQG. Moreover, LQR & LQG settings are MDP & POMDP models with the state, action,
reward spaces not constrained to be bounded. They are fundamental models in MDPs and POMDPs:

• LQR & LQG are among few continuous settings where the optimal policies exist (and mainly have closed form) [1].
• For the general model of $x_{t+1} = f(x_t, u_t, w_t)$ with Gaussian $w_t$, using representation theory results (e.g. Koopman
theory or RKHS theory) any such $f$ can be written as linear function of its basis. Thus, MDPs or POMDPs can be
written as LDS up to some considerations and a further change of bases. These principles have been intuitively used
in deep RL (e.g. Zhang et al. "SOLAR: Deep Structured Representations for Model-Based RL" 2019).

Based on these facts, the study of LDS, with LQG being one of the most challenging ones, is an important problem
in RL. Note that prior works in this area, such as [6,12,37,40-46] have been published in recent machine learning
conferences (NeurIPS, ICML...). Therefore, we do not see why this paper would be less relevant to our community.

**Regret without the persistence of excitation (PE):** (**R1**, **R2**). In general, PE is standard in control theory since it
allows asymptotic convergence of algorithms involving system identification, adaptive prediction, and control (e.g.
Boyd & Sastry, *On Parameter Convergence in Adaptive Control*, 1983; Green & Moore, *Persistence of excitation in*
*linear system*, 1986). If PE is absent, we provide two general algorithms stated in Cor. 6.2 and H.1:

1. The agent uses a warm-up period of $O(\sqrt{T})$ after which it commits to a controller yielding a regret of $\sqrt{T}$.
2. This approach is concerned more with adaptive model estimation than regret minimization. The agent adds Gaussian
noise to the control input which yields regret of $T^{2/3}$, while adaptively improving the accuracy of model estimates.

**R1**: We are delighted to hear the kind words of R1 about our novel results.
**PE policies:** This is a mild condition and most of the well-known controllers ($H_2, H_\infty$) satisfy it. For example, consider
a unary DFC $\mathbf{M}$ and the PE condition given in Appendix E.2. For $\mathbf{M}$ to be not PE, we need to have an extremely wide
matrix of $p \times O(\bar{H}(n+m))$ dimensions (block row of the matrix that maps past noise to input) to be row rank deficient,
where $\bar{H}$ is our choice and a large number. Thus, it is quite hard and pathological to design an LQG with a small
neighborhood such that there exists an $\mathbf{M}$ which is not PE for the models in the neighborhood. Moreover, if $\mathbf{M}_\star$ is PE,
then there is a neighborhood around $\mathbf{M}_\star$ consisting of all PE controllers. Note that the prior works also rely on PE for
consistent estimates. We appreciate R1 for bringing up this discussion. Per R1's suggestion, we added the rigorous
version of the above discussion and explanation to the main text. Relaxing the PE requirement is still an important open
problem which we will discuss in the conclusion. In light of this, we would like to invite R1 to increase their score.

**R2**: We thank R2 for their insightful comments about our novel results in the field of online control of LDS.
**Regarding stabilizing controller:** The majority of the prior works in partially observable LDS consider stable systems
[3-5,9,12]. Recently, [6] made a significant effort to generalize aspects of this to stabilizable systems when a stabilizing
controller is given. Note that, many partially observed systems cannot be stabilized by a static feedback controller and
the assumption of the existence of such controller is somewhat restrictive (see Halevi *Stable LQG controllers* 1994). In
the current paper, we provide a general framework of learning and regret analysis in partially obsevable stable LDS, and
avoid further complications to convey this core contribution.
At this point, we would appreciate if R1 & R2 could convince R3 of the importance of partially observable LDS in RL.

**R3**: As we describe in the first paragraph of the rebuttal, we propose the **first finite-time closed-loop learning**
**algorithm of partially observable LDS**. Prior estimation methods only work when actions are random iid noises. We
strongly believe that this result alone is a significant contribution to the field. Building on the mentioned estimation
method, we use online learning techniques [11] for the final step of controller learning which are also used in [6]. We
adapt the controller design of [6] to our formulation of system identification. We derive novel sample complexity
requirements to satisfy closed-loop stability and persistence of excitation during the adaptive control period.

[Meta-Review · NeurIPS 2020]

In discussion the reviewers felt that the main result of the paper---that logarithmic regret is possible for LQG under sufficient observation noise---is significant and worth pointing out, especially given \sqrt{T} lower bounds for the fully observable setting. The reviewers did feel that the framing of the results can be improved, and I encourage the authors to do this for the final version. In particular 1) the result is not necessarily surprising given the noise assumptions, and it would be good to be more transparent about this, and 2) the claim (which is even present in the rebuttal) that the exploration scheme here is "strategic" in some way compared to prior results based on injecting random noise is very questionable, and it is indeed not clear that the techniques here can be extended beyond linear control.